# Targeting HDAC6 to treat heart failure with preserved ejection fraction in mice

Sara Ranjbarvaziri[1,2], Aliya Zeng[1,2], Iris Wu [1], Amara Greer-Short[1], Farshad Farshidfar[1], Ana Budan[1], Emma Xu[1], Reva Shenwai[1], Matthew Kozubov[1], Cindy Li[1], Melissa Van Pell[1], Francis Grafton[1], Charles E MacKay[1], Xiaomei Song[1], James R Priest[1], Gretchen Argast[1], Mohammad A. Mandegar[1], Timothy Hoey[1] & Jin Yang [1] ✉

Heart failure with preserved ejection fraction (HFpEF) poses therapeutic challenges due to the limited treatment options. Building upon our previous research that demonstrates the efficacy of histone deacetylase 6 (HDAC6) inhibition in a genetic cardiomyopathy model, we investigate HDAC6's role in HFpEF due to their shared mechanisms of inflammation and metabolism. Here, we show that inhibiting HDAC6 with TYA-018 effectively reverses established heart failure and its associated symptoms in male HFpEF mouse models. Additionally, in male mice lacking *Hdac6* gene, HFpEF progression is delayed and they are resistant to TYA-018's effects. The efficacy of TYA-018 is comparable to a sodium-glucose cotransporter 2 (SGLT2) inhibitor, and the combination shows enhanced effects. Mechanistically, TYA-018 restores gene expression related to hypertrophy, fibrosis, and mitochondrial energy production in HFpEF heart tissues. Furthermore, TYA-018 also inhibits activation of human cardiac fibroblasts and enhances mitochondrial respiratory capacity in cardiomyocytes. In this work, our findings show that HDAC6 impacts on heart pathophysiology and is a promising target for HFpEF treatment.

Heart failure with preserved ejection fraction (HFpEF) accounts for more than 50% of all HF cases and is associated with high morbidity and mortality; yet, there are limited disease-modifying options that specifically treat HFpEF. For example, sodium-glucose transport protein 2 (SGLT2) inhibitors and angiotensin receptor-neprilysin inhibitors have shown clinical evidence of reducing hospitalization for HF, but their benefits are generally limited to patients with HFpEF who have a lower EF. These therapies also require careful dosing and monitoring to reduce the risk of adverse events, such as hypotension and low renal function[1,2]. To develop more effective therapies for HFpEF, we need to better understand the molecular mechanisms governing HFpEF.

HFpEF is characterized by diastolic dysfunction with a normal EF, a nondilated left ventricle with concentric remodeling, or left ventricular (LV) hypertrophy. LV diastolic stiffness and relaxation are also impaired, resulting in higher LV filling pressures. These characteristics contribute to symptoms of dyspnea, abnormalities in lung gas

exchange, impaired aerobic capacity, and the development of pulmonary hypertension[1,3].

In HFpEF, the severity of diastolic dysfunction is influenced by myocardial extracellular matrix remodeling and mitochondrial disorder[4–6]. The collagen deposited by myofibroblasts disrupts the electromechanical coordination between cardiomyocytes, leading to ventricular stiffness, impaired diastolic relaxation, and filling[7]. Also, HFpEF is characterized by a compromised energy metabolism, leading to a deficit in energy production. Experimental HFpEF model is associated with perturbations in fatty acid oxidation, redox reactions, and the synthesis of adenosine triphosphate in the mitochondria[8].

Preclinical studies have demonstrated the cardioprotective effects of a pan-HDAC inhibitor (Givinostat) in mouse and SAHA in a feline model of diastolic dysfunction[9,10]. Previously, we found that inhibiting HDAC6 altered mechanisms that contribute to dilated cardiomyopathy[11]. Specifically, TYA-018, a highly selective and orally

---

[1]Tenaya Therapeutics, South San Francisco, CA, USA. [2]These authors contributed equally: Sara Ranjbarvaziri, Aliya Zeng. ✉e-mail: flairjinyang@gmail.com

available HDAC6 inhibitor, increased the expression of targets associated with fatty acid metabolism, protein metabolism, and oxidative phosphorylation in a mouse model of dilated cardiomyopathy[11]. Given the involvement of these mechanisms in pathophysiological processes associated with HFpEF[12], we hypothesized that HDAC6 might play a significant role in HFpEF as well.

In this study, we developed a mouse model induced by a high-fat diet (HFD) in combination with moderate transverse aortic constriction (mTAC) to mimics crucial hemodynamic characteristics of HFpEF observed in patients. Utilizing this model and another HFD + L-NAME model[13], we investigate the impact of inhibiting HDAC6−either with a small molecule or by genetic deletion−on HFpEF and its underlying mechanisms. Our results support the direct role of HDAC6 on HFpEF pathophysiology within the heart, suggesting that inhibiting HDAC6 could be a promising approach for treating HFpEF.

## Results

### Develop a mouse model recapitulates human HFpEF

Age, hypertension, obesity and metabolic syndrome are important risk factors in HFpEF pathophysiology[14,15]. In this study, we took a "two-hit" approach using metabolic stress induced by an HFD and mechanical stress induced by chronic moderate pressure overload in the heart with mTAC to mimic the physiological and morphological features seen in patients with HFpEF. We divided 2-month-old male C57BL/6 J mice into four groups with one of the following 16-week regimens: (1) HFD (60% kilocalories from fat); (2) mTAC (with a standard chow diet); (3) HFD+mTAC; or (4) controls (no mTAC and a standard chow diet) (Fig. 1a). In these mice, mTAC generated moderate pressure overload to the left ventricle (Supplementary Fig. 1a and 1b), and an HFD led to increased body weight and glucose intolerance (Fig. 1b and Supplementary Fig. 1c and 1d).

To evaluate heart function in these mice, we used longitudinal echocardiography, Doppler imaging, and invasive catheterization. We found that compared to controls, HFD+mTAC mice had a persistently preserved left ventricular ejection fraction (LVEF), whereas HFD and mTAC mice had a slightly lower LVEF. HFD+mTAC mice also had significant concentric LV hypertrophy, indicated by higher global hypertrophy (LV mass) and LV posterior wall thickness at diastole (LVPWd) (Fig. 1c−e and Supplementary Fig. 1e-i). Although all groups showed varying degrees of diastolic dysfunction, HFD+mTAC mice demonstrated signs of increased LV filling pressure as measured with noninvasive Doppler imaging and invasive catheterization 16 weeks after model induction (Fig. 1f−i, Supplementary Fig. 1j, k). It is worth noting that mice subjected to the HFD+mTAC treatment exhibited an ~20% mortality rate within a 16-week period, while no mortality was observed in the other groups (Supplementary Fig. 1l).

A key feature of HFpEF is exercise intolerance, which contributes to reduced quality of life. Thus, we evaluated exercise performance in each group of mice 16 weeks after induction. Although HFD mice gained weight and had lower exercise performance, HFD+mTAC mice ran a significantly shorter distance than HFD alone group, consistent with their pronounced diastolic dysfunction (Fig. 1j). These findings support that HFD+mTAC mice recapitulated the pathological phenotypes seen in patients with HFpEF.

### Hearts of HFD + mTAC mice show HFpEF transcriptomic signatures

To characterize the transcriptional profile of our HFpEF model, we completed bulk RNA sequencing and unbiased Gene Set Enrichment Analysis (GSEA) of whole LV homogenates from control and HFpEF mice. Several gene clusters were enriched in HFpEF hearts, including genes associated with muscle hypertrophy and contraction, fibrosis [transforming growth factor β receptor signaling (TGFβ), extracellular matrix structural constituent], and platelet-derived growth factor receptor (PDGFR) signaling (Fig. 2a). Conversely, some genes were depleted in HFpEF hearts, including gene sets associated with mitochondrial function (electron transport chain, oxidative phosphorylation) (Fig. 2a). These gene expression profiles agree with transcriptomic signatures of pathophysiological components that underlie human HFpEF[16,17], suggesting the translational utility of the HFD+mTAC model.

### HDAC6 upregulation in HFpEF hearts

Previous research has demonstrated the beneficial effects of inhibiting HDAC6 in a genetic cardiomyopathy model[11]. Given the shared underlying mechanisms involving inflammation and metabolism in both cardiomyopathy and HFpEF, we were driven to explore the potential role of HDAC6 in HFpEF. We quantified HDAC6 protein level in heart tissues of HFD+mTAC mice. In these mice, HDAC6 protein was significantly upregulated compared to controls (Fig. 2b). Remarkably, cardiac HDAC6 showed a positive correlation with marker genes associated with heart function (*Nppb*) and fibrosis (*Col3a1*) (Supplementary Fig. 2a, b).

Next, we investigated whether HDAC6 exhibits a conserved mechanism in HFpEF using a published mouse model of HFpEF. In this model, mice are fed an HFD and treated with L-NAME (L-N$^G$-Nitro arginine methyl ester) for 20 weeks[13]. These HFD + L-NAME mice developed cardiac hypertrophy and diastolic dysfunction with preserved EF (Supplementary Fig. 2c−f). In heart tissues from these mice, the marker genes *Nppb* and *Col3a1* mRNA were significantly upregulated, supporting the establishment of HFpEF in this model (Supplementary Fig. 2g, h). In HFD + L-NAME mice, the expression of HDAC6 protein was also increased (Fig. 2c), correlated with the upregulation of *Nppb* and *Col3a1*, and severity of diastolic dysfunction [based on the early diastolic velocity ratio (E/e′)] and cardiac hypertrophy (based on LVPWd) (Supplementary Fig. 2i, j). In these cases, a higher level of HDAC6 protein expression was associated with more severe HF. This correlation suggests that HDAC6 plays a role in the development of HFpEF (Fig. 2c, Supplementary Fig. 2k−l). We also assessed HDAC6 enzymatic activity by quantifying the acetylation status of its widely recognized substrate, tubulin, in cardiac tissue samples collected from the two HFpEF models. Interestingly, the extent of tubulin acetylation exhibited a consistent pattern across both the control group and the HFpEF group (Supplementary Fig. 2m, n).

To test whether HDAC6 is also upregulated in human hearts with HFpEF, we analyzed publicly available RNA-Seq data of myocardial biopsies from control and failing human hearts[16]. HDAC6 mRNA levels were trending increased (+7.4%) in human hearts with HFpEF versus controls, although this difference is not statistically significant (Fig. 2d). Future studies should incorporate more data points to achieve a robust statistical comparison. Furthermore, given protein levels can provide additional insights into the functional status of HDAC6 beyond mRNA expression, it is crucial to analyze HDAC6 protein levels in HFpEF patient hearts, as have done in the mouse hearts of the two HFpEF models (Fig. 2b and 2c).

### HDAC6 inhibition improves HFpEF heart function

To determine how HDAC6 contributes to HFpEF, we utilized the small molecule TYA-018 to inhibit HDAC6 activity in HFD+mTAC mice[11]. We treated mice with TYA-018 (15 mg/kg) or vehicle (administered orally once per day) starting 16 weeks after mTAC and HFD induction. At this time, HFD+mTAC mice had established HFpEF phenotypes shown by cardiac hypertrophy and diastolic dysfunction with preserved EF (Fig. 3a and Supplementary Fig. 3a−f). Then, 3 to 6 weeks after starting TYA-018 treatment at 16 weeks, we assessed LV structure and heart function with echocardiography.

Six weeks after starting treatment, the vehicle-treated mice with HFD+mTAC developed severe cardiac hypertrophy and diastolic dysfunction, while the TYA-018-treated mice showed continual improvements in left ventricular (LV) remodeling and diastolic function (Supplementary Fig. 3a−h). The TYA-018-treated mice showed declining LVPWd with no effects on EF, and improved diastolic function

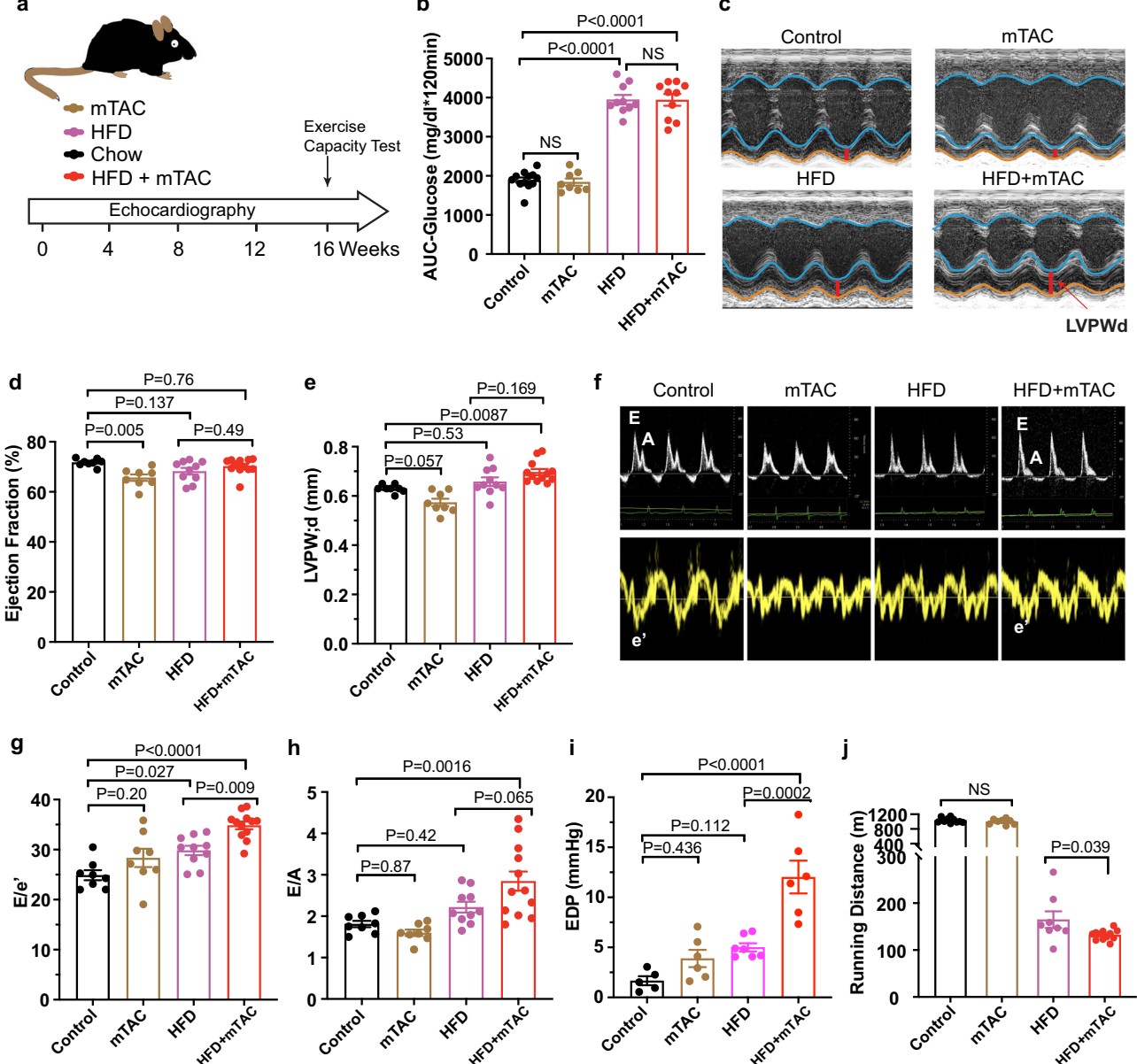

**Fig. 1 | HFD + mTAC mice developed key phenotypes of clinical HFpEF.**
**a** Schematic overview of study design. Wild-type male C57BL/6J mice were separated into the following regimens for 16 weeks: standard chow (control), HFD, mTAC, or HFD+mTAC for 16 weeks. Heart function was measured with echocardiography every 4 weeks. **b** Area under the curve (AUC) of an intraperitoneal glucose-tolerance test of mice 16 weeks after induction (control $n = 12$, mTAC $n = 8$, HFD $n = 10$, HFD+mTAC $n = 10$ mice). **c** Representative LV M-mode echocardiographic tracings. Images are representative of 8–10 independent mice.
**d**, **e** Echocardiographic measurement of ejection fraction and LVPWd 16 weeks after induction (control $n = 8$, mTAC $n = 8$, HFD $n = 10$, HFD+mTAC $n = 12$ mice).
**f** Representative pulsed-wave Doppler (top) and tissue Doppler (bottom) tracings after 16 weeks of induction. Images are representative of 8–10 independent mice.

**g**, **h** Non-invasive Doppler analysis of E/e' and E/A ratios 16 weeks after induction (control $n = 8$, mTAC $n = 8$, HFD $n = 10$, HFD+mTAC $n = 12$ mice). **i** Pressure-volume loop measurement of EDP in each group of mice at 16 weeks after induction (control $n = 5$, mTAC $n = 6$, HFD $n = 7$, HFD+mTAC $n = 6$ mice). **j** Running distance during exercise exhaustion test (control $n = 10$, mTAC $n = 8$, HFD $n = 8$, HFD+mTAC $n = 12$ mice). Data are expressed as the mean ± SEM. Statistical analysis was performed using one-way ANOVA followed by Tukey's multiple comparisons test (**b**, **d**, **e**, **g**–**i**). Statistical significance was assessed by unpaired two-sided Student's $t$ test between the two groups of HFD and HFD+mTAC (**j**). The exact $P$ values are shown in the figures. NS, not significant. Source data are provided as a Source Data file.

parameters such as normalized IVRT, E/e', and E/A (Fig. 3b–g and Supplementary Fig. 3a–h). This improvement was confirmed by measuring end-diastolic pressure with an invasive micro-admittance catheter, which showed that TYA-018 treatment resulted in normalized end-diastolic pressure (Fig. 3h). The findings suggest that inhibiting HDAC6 has the potential to reverse preexisting diastolic abnormalities and improve cardiac relaxation and blood filling in HFpEF, leading to continual improvement in LV remodeling and diastolic function over time.

The improved cardiac phenotype in HFD+mTAC mice treated with TYA-018 was accompanied by reduced lung weight, a measure of pulmonary congestion and a clinical sign of HF in rodent models (Fig. 3i). These mice also had a lower heart weight, consistent with reduced hypertrophy (Fig. 3j). HFD+mTAC mice treated with TYA-018 also had a significantly better exercise performance (when they did not show a change in body weight) than those treated with vehicle (Fig. 3k–l and Supplementary Fig. 3i, j). These mice also showed improved glucose tolerance and restored fasting glucose (Fig. 3m–o).

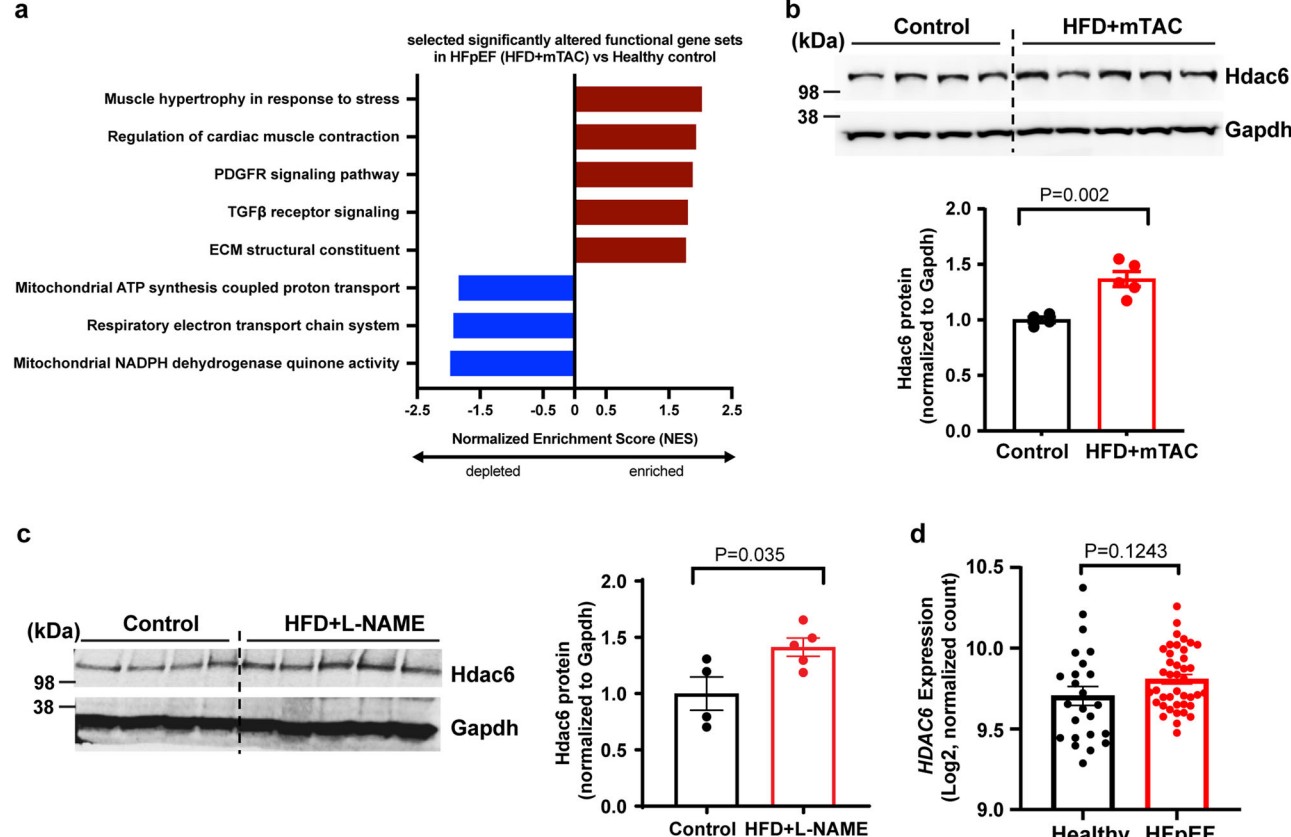

**Fig. 2 | HDAC6 upregulation in heart tissues from HFpEF mouse models and patients with HFpEF. a** Gene Ontology terms associated with upregulated and downregulated genes in HFD+mTAC mice ($n = 5$) versus control mice ($n = 3$). **b** Western blot analysis and quantitation of Hdac6 protein in heart tissues from control ($n = 4$) and HFD+mTAC mice ($n = 5$). **c** Western blot analysis and quantitation of Hdac6 protein in heart tissues of control ($n = 4$) and HFD + L-NAME mice ($n = 5$). **d** Quantitation of *HDAC6* mRNA expression in human heart tissues of healthy ($n = 24$) and patients with HFpEF ($n = 41$) from published RNA-Seq data. Data are expressed as the mean ± SEM. Statistical significance was assessed by unpaired two-sided Student's t test (**b**, **c**), or unpaired two-sided t test with Welch's correction (**d**). The exact *P* values are shown in the figure. ECM, extracellular matrix. Source data are provided as a Source Data file.

Notably, no adverse effects of TYA-018 treatment were observed during the study period. Additionally, TYA-018 treatment increased acetylated tubulin levels in HFD+mTAC mice, indicating robust HDAC6 inhibition and target engagement (Supplementary Fig. 2m). Collectively, these data show that inhibiting HDAC6 with TYA-018 rescues diastolic dysfunction and alleviates signs of HF in HFpEF mice, suggesting that HDAC6 activation contributes to HFpEF pathogenesis.

### Genetic HDAC6 deletion slows HFpEF progression

To further assess the role of HDAC6 in HFpEF, we employed a genetic approach and studied *Hdac6* knockout mice alongside their wild-type littermates. Both groups were subjected to a high-fat diet supplemented with L-NAME (Fig. 4a, Supplementary Fig. 4a). At 16 weeks, both groups exhibited similar levels of obesity (Supplementary Fig. 4b). However, *Hdac6* knockout mice showed a decelerated disease progression, as evidenced by reduced LV Mass, LVPWd, IVRT and E/e' ratio at both 8 weeks and 12 weeks (Fig. 4b–d and Supplementary Fig. 4c, d). It is worth noting that there were no statistically significant differences between WT and *Hdac6* knockout mice in these parameters at 16 weeks. These findings support the hypothesis that HDAC6 plays a role in the development of HFpEF.

### TYA-018 targets HDAC6 for heart improvement

Given that both *Hdac6* knockout mice and their littermates developed HFpEF phenotypes after 16 weeks on HFD + L-NAME, our next aim was to evaluate the clinical utility of TYA-018 and determine its specificity

on targeting HDAC6 to improve HFpEF. *Hdac6* knockout mice and their littermates were randomly to receive either TYA-018 or vehicle for 8 weeks (Fig. 4a). Remarkably, treatment with TYA-018 for 8 weeks significantly attenuated diastolic dysfunction in wild-type mice (Fig. 4e–g). However, this therapeutic efficacy was not observed in *Hdac6* knockout mice following the HFD + L-NAME regimen (Fig. 4e–g and Supplementary Fig. 4e–i). These data provide further support that HDAC6 serves as a target for HFpEF treatment and highlight the inhibitory effect of TYA-018 on HDAC6, leading to a notable improvement in heart function in HFpEF.

### HDAC6 inhibition shows comparable efficacy to empagliflozin

To ensure the clinical utility and translatability of TYA-018, we conducted a side-by-side evaluation of its efficacy with empagliflozin, a selective inhibitor of SGLT2 that has been approved by the Food and Drug Administration (FDA) for treating patients with HFpEF[18]. Mice with established HFpEF, induced by HFD + L-NAME, were orally treated with TYA-018, empagliflozin, or vehicle control once daily for 9 weeks using optimal doses (Fig. 5a). A single dose of TYA-018 reduced fasting glucose and improved glucose tolerance to similar levels as empagliflozin (Fig. 5b–d). Both TYA-018 and empagliflozin effectively reduced LV hypertrophy and improved diastolic function, while not affecting EF, heart rate, and blood pressure (Fig. 5e–i and Supplementary Fig. 5a–g). TYA-018 also showed superior activity in comparison to empagliflozin in improving diastolic function, as measured by the reduction in E/e'. Furthermore, TYA-018 demonstrated greater reductions in marker genes associated with heart failure (HF) and fibrosis, further

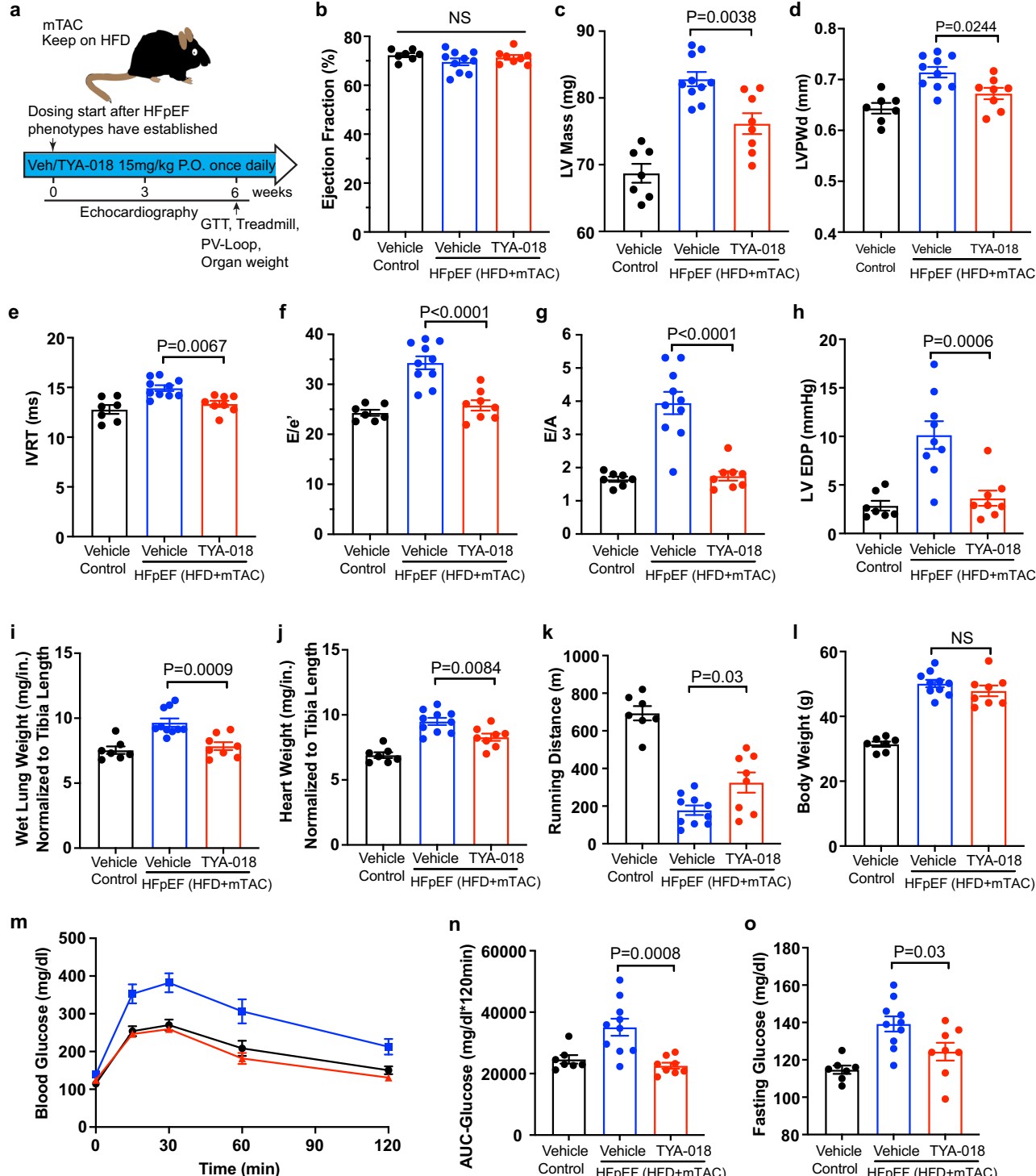

**Fig. 3 | HDAC6 inhibition by TYA-018 reverses pre-existing heart failure in HFpEF model. a** Schematic overview of study design. C57BL/6J mice with established HFpEF induced by HFD+mTAC were randomized to receive oral dosing of vehicle or TYA-018 (15 mg/kg) once per day. Heart function was measured with echocardiography at 3 and 6 weeks after treatment began. Quantitation of (**b**) EF, (**c**) LV mass and (**d**) LVPWd by echocardiography after 6 weeks of treatment. Quantitation of (**e**) IVRT, (**f**) E/e' ratio, and (**g**) E/A ratio by non-invasive Doppler analysis after 6 weeks of treatment. **h** EDP invasively measured by intracardiac catheterization (Control *n* = 7, HFD+mTAC-Vehicle *n* = 9, HFD+mTAC-TYA-018 *n* = 8 mice). **i** Lung and (**j**) heart weight normalized to tibia length. **k** Running distance during exercise exhaustion test after 6 weeks of treatment. **l** Body weight of mice. **m** Blood glucose and (**n**) quantification of the area under the curve (AUC) of intraperitoneal glucose-tolerance test (GTT) after TYA-018 treatment. **o** Fasting blood glucose at 6 h after the last dose of TYA-018. Mice numbers: Control *n* = 7, HFD+mTAC-vehicle *n* = 10, HFD+mTAC-TYA-018 *n* = 8 (**b**–**g**, **i**–**o**). Data are expressed as the mean ± SEM. Statistical analysis was performed using one-way ANOVA followed by Tukey's multiple comparisons test (**b**–**l**, **n**, **o**). The exact *P* values are shown in the figures. NS not significant. Source data are provided as a Source Data file.

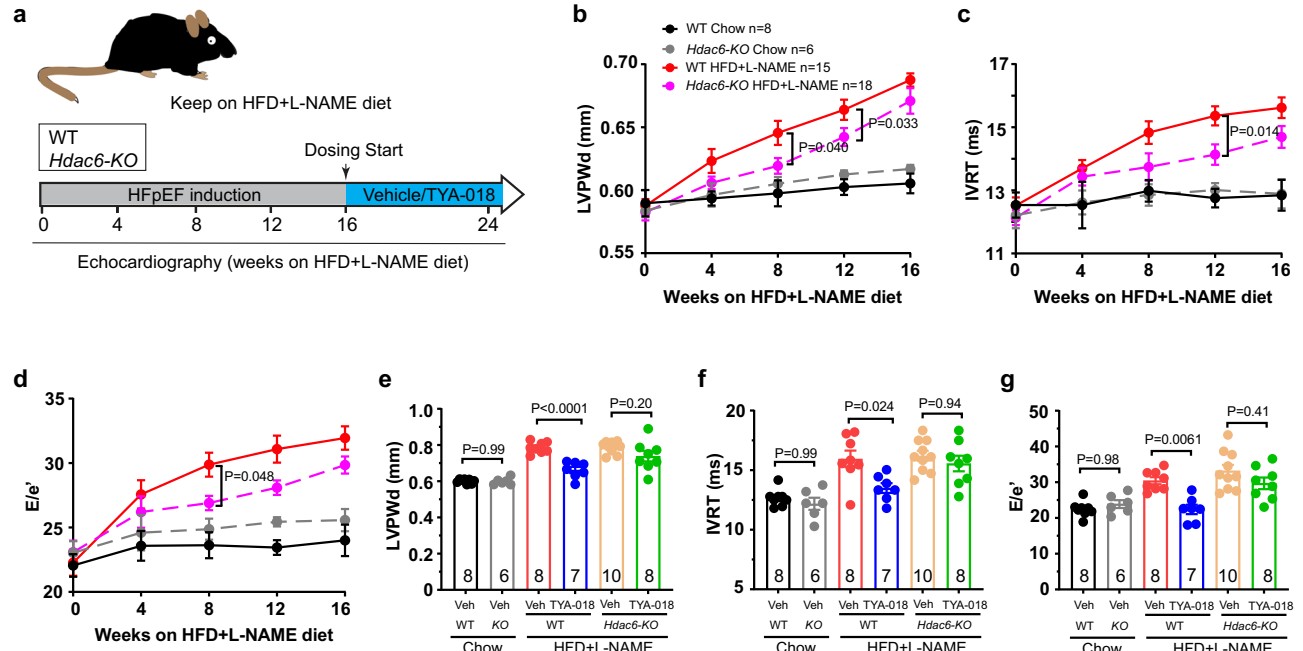

**Fig. 4 | *Hdac6-KO* mice exhibit slower progression of HFpEF, and do not respond to TYA-018 treatment. a** Schematic overview of study design. *Hdac6* knockout (KO) and wild-type (WT) mice treated with HFD + L-NAME for 16 weeks and then randomized to receive oral dosing of vehicle or TYA-018 (15 mg/kg) once per day for 8 weeks. Heart function was measured by echocardiography. Time course of (**b**) LVPWd, (**c**) IVRT (**d**) E/e′ ratio at 4, 8, 12, and 16 weeks after HFD + L-

NAME induction. Quantitation of (**e**) LVPWd, (**f**) IVRT and (**g**) E/e′ ratio after 8 weeks of TYA-018 treatment. Mice number in each group is indicated in graphs. Data are expressed as the mean ± SEM. Statistical analysis was performed using two-way ANOVA (**b–d**) or one-way ANOVA followed by Tukey's multiple comparisons test (**e–g**). The exact *P* values are shown in the figures. Source data are provided as a Source Data file.

highlighting its distinct mechanism of activity (Fig. 5j, k). The efficacy of empagliflozin in this experiment supports the validity of the HFD + L-NAME HFpEF model, while these results, along with the direct and systemic benefits observed in other preclinical studies, suggest the potential translatability of TYA-018's efficacy to clinical development.

## HDAC6 inhibition dampens HFpEF response, boosts mitochondrial energetics

Next, we processed heart tissues for untargeted transcriptional profiling of protein-coding genes with bulk RNA-Seq. To evaluate functional perturbations, we used unbiased GSEA[19]. We found that gene sets linked to inflammation, fibrosis, and oxidative stress were enriched in vehicle-treated HFpEF mice, whereas HFpEF mice treated with TYA-018 exhibited a reversal of these transcriptional changes (Fig. 6a). Conversely, gene sets associated with mitochondrial function (e.g., oxidative phosphorylation, mitochondrial biogenesis, different metabolic pathways) were significantly depleted in vehicle-treated HFpEF mice. In contrast, HFpEF mice treated with TYA-018 showed a correction of these transcriptional signatures (Fig. 6a). When compared to empagliflozin, TYA-018 showed greater or similar effects on correcting expression of these gene sets in HFpEF. Furthermore, the differential expression analysis of selected genes from the gene set terms revealed a trending correction of key genes associated with fibrosis and mitochondrial biogenesis in HFpEF mice following TYA-018 treatment (Fig. 6b).

To test whether these transcriptional changes are associated with cardiac function, we performed Pearson correlation coefficient analysis between the expression of genes identified by GSEA analysis and the key parameters of LV diastolic function, LV hypertrophy, E/e′, and LVPWd. TYA-018 and empagliflozin both significantly reduced expression of several genes associated with fibrosis (*Lum, Mmp2, Mmp17, Timp1,* and *Pdgfra*). The expression of these genes closely correlated with improvements in E/e′ and LVPWd (Supplementary Fig. 5h). In contrast, both TYA-018 and empagliflozin significantly

increased the expression of genes associated with different subunits of the mitochondrial respiratory electron transport chain (*Atp5h, Atp5o, Atp5j,* and *Ndufa5*). These gene expressions also correlated with E/e′ and LVPWd (Supplementary Fig. 5i).

To understand the effects of TYA-018 and empagliflozin on cardiac cells in HFpEF model, we completed single-nuclear RNA sequencing (snRNA-seq) using the 10X Genomics platform (Fig. 5a). The snRNA-seq data was analyzed using unsupervised clustering, which identified several populations of cardiac cells including cardiomyocytes, fibroblasts, endothelial cells, pericytes, and immune cells (Fig. 6c). Cardiomyocytes and cardiac fibroblasts are the main cell types in the myocardium, and their dysfunction is the hallmark of HFpEF[20]. To identify changes in the molecular signatures of the two cardiac cell types, nuclei of all samples were computationally ordered by trajectory analysis. The cardiomyocytes were divided into four distinct subclusters, which revealed that cluster 3 showed a greater enrichment in the HFpEF model compared to controls (Fig. 6d, f, h). Some of the top genes that define CM_3 are well-known genes associated with cardiac stress, including *Abra, Atf3, Ccn1, Nppb,* and *Xirp2*[21–24] (Fig. 6f, h). Analysis of the genes associated with this cluster showed a significant involvement of pathways related to TNFa signaling, apoptosis, unfolded protein response, P53 pathway, and others (Fig. 6i). On the other hand, the fibroblasts were divided into nine subclusters and showed a large enrichment of cluster 6 in the HFpEF model (Fig. 6e, g and j). This cluster was characterized by genes associated with the GO terms epithelial-mesenchymal transition and TGFb signaling (Fig. 6k). Notably, treatment with TYA-018 or empagliflozin reversed the enrichments of cardiomyocyte cluster 3 and fibroblast cluster 6 back to control amounts (Fig. 6h, j), suggesting that these two cell types are the primary targets of these treatments in the HFpEF model. More importantly, these treatments led to the normalization of a cell population (CM_3) that expressing cardiac stress markers, consistent with the complete rescue of diastolic function. Overall, these data provide insights into the effects of TYA-018 and

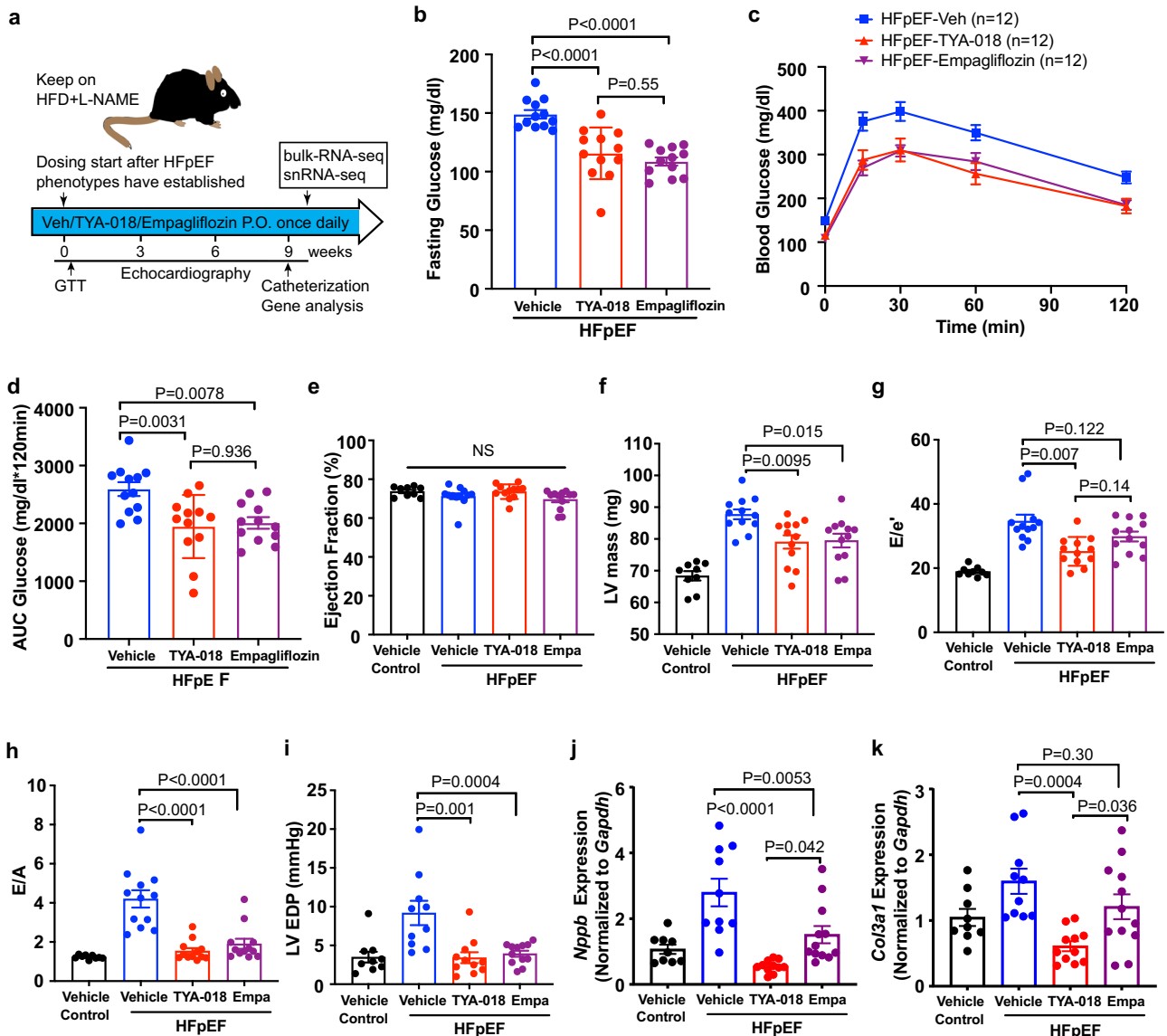

**Fig. 5 | TYA-018 has comparable efficacy to empagliflozin in HFpEF model.**
**a** Schematic overview of study design. C57BL/6J mice with established HFpEF induced by HFD + L-NAME were randomized to receive oral dosing of vehicle, TYA-018 (15 mg/kg), or empagliflozin (10 mg/kg) once per day. **b** Quantitation of blood glucose at 6 h after dosing with TYA-018 or empagliflozin and fasting. Blood glucose (**c**) and area under the curve (**d**) of the intraperitoneal glucose-tolerance test (GTT) (*n* = 12 mice in each group). Heart function was measured with echocardiography every 3 weeks, and mice were euthanized after 9 weeks of treatment. Quantitation of (**e**) EF, (**f**) LV mass, (**g**) E/e' and (**h**) E/A ratios by echocardiography and non-invasive Doppler after 9 weeks of treatment (*n* = 9 in control vehicle group,

*n* = 12 in each of the other groups). Quantitation of EDP (**i**) by invasive intracardiac catheterization before euthanization at 9 weeks after treatment (Vehicle control *n* = 9, HFpEF-vehicle *n* = 10, HFpEF-TYA-018 *n* = 11, HFpEF-Empa *n* = 12). Quantitation of *Nppb* (**j**) and *Col3a1* (**k**) mRNA in mouse hearts after 9 weeks of treatment (Vehicle control *n* = 9, HFpEF-vehicle *n* = 10, HFpEF-TYA-018 *n* = 11, HFpEF-Empa *n* = 12. Data are expressed as the mean ± SEM. Statistical analysis was performed using one-way ANOVA followed by Tukey's multiple comparisons test (**b**, **d–k**). The exact *P* values are shown in the figures. NS not significant. Source data are provided as a Source Data file.

empagliflozin on cardiac cells in HFpEF and highlight the potential importance of these treatments in improving heart function.

Next, we aimed to decipher if these gene sets are involved in the mechanism of TYA-018 in HFpEF. We performed RNA-Seq analysis of heart tissues harvested 6 h after one dose of vehicle or TYA-018 (15 mg/kg), before heart structure and function showed any improvements. Similar to long-term treatment with TYA-018, a single dose of TYA-018 corrected gene sets associated with fibrosis and inflammation. This single dose also significantly enriched gene sets involved in mitochondrial function and metabolism (Supplementary Fig. 6a, b). These data suggest that TYA-018 regulates genes associated with fibrosis and mitochondrial function in HFpEF.

## Combination effects of TYA-018 and empagliflozin in HFpEF

Given the distinct mechanisms of action between TYA-018 and empagliflozin, we hypothesized that they may have additive or synergistic effects when used in combination. A combination of low doses of TYA-018 (0.3 mg/kg) and empagliflozin (0.5 mg/kg) was administered orally to mice with established HFpEF. Other groups of mice were treated with single compounds or vehicle. After eight weeks of treatment, the combination therapy resulted in continual improvements in LV remodeling and diastolic function, as well as reductions in fasting glucose (Supplementary Fig. 7a–n). These improvements were not observed or less extensive in mice treated with either compound alone or with the vehicle, suggesting that the combination therapy may have a greater impact on the treatment of

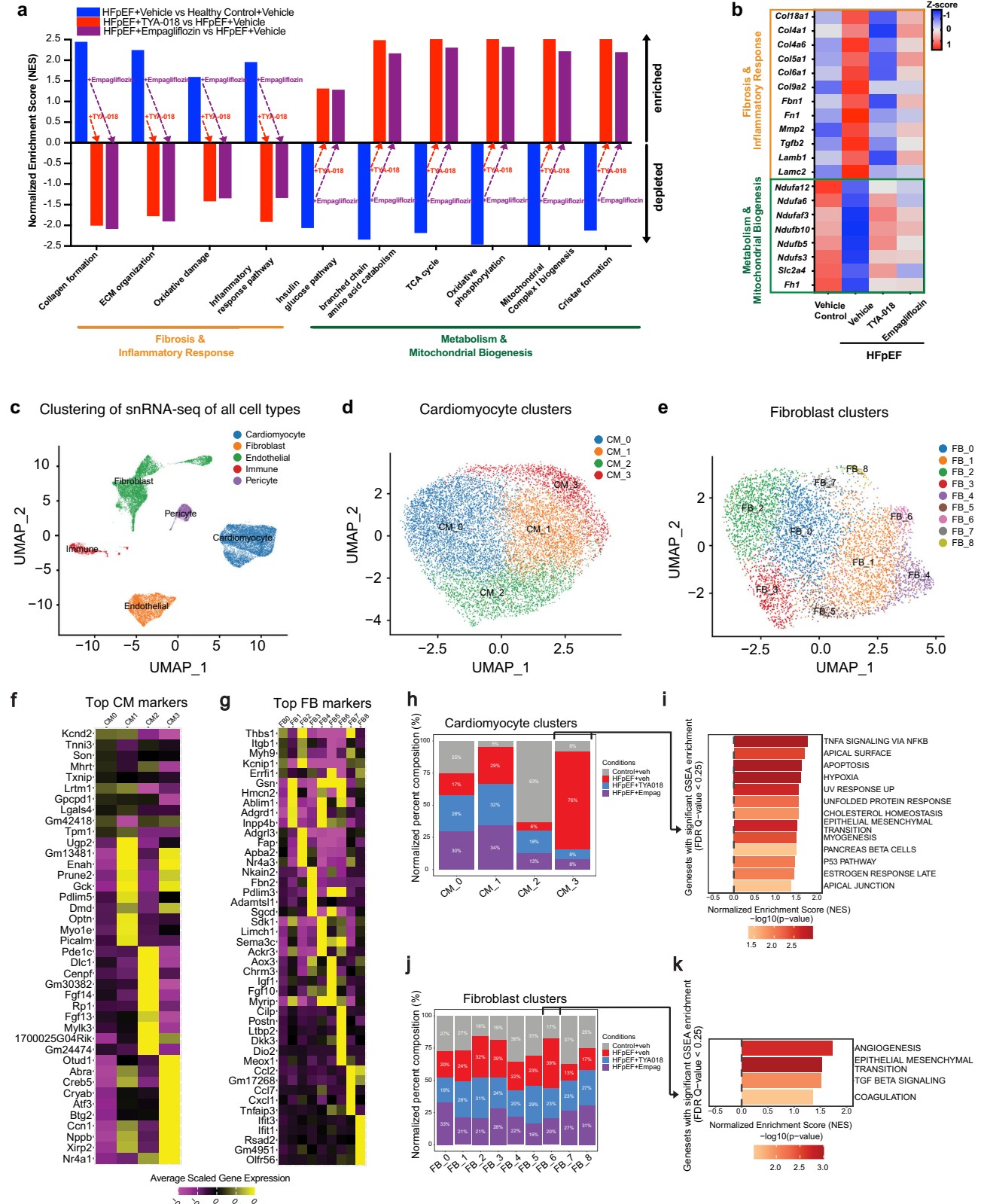

**Fig. 6 | HDAC6 inhibition by TYA-018 restores dysregulated fibrosis and metabolic transcripts in heart tissue from HFpEF mice. a** Significantly altered gene sets from MSigDB canonical pathways. **b** Heatmap of differentially regulated genes associated with cardiac fibrosis as well as inflammatory and mitochondrial function in all groups (*n* = 8–11 mice per group). **c** Uniform manifold approximation and projection (UMAP) plot after single-nuclei RNA sequencing of left ventricular tissue. **d** UMAP plot of four cardiomyocytes (CM) subclusters. **e** UMAP plot of nine

fibroblast (FB) subclusters. **f** Heatmap of the top 10 maker genes for each CM subclusters. **g** Heat map showing the expression of the top 5 maker genes for each FB subclusters. **h** Composition of the four CM subclusters within the four groups. **i** Gene ontology enrichment analysis of CM cluster 3. **j** Composition of the nine FB subclusters within the four groups. **k** Gene ontology enrichment analysis of FB cluster 6. ECM extracellular matrix; TCA tricarboxylic acid.

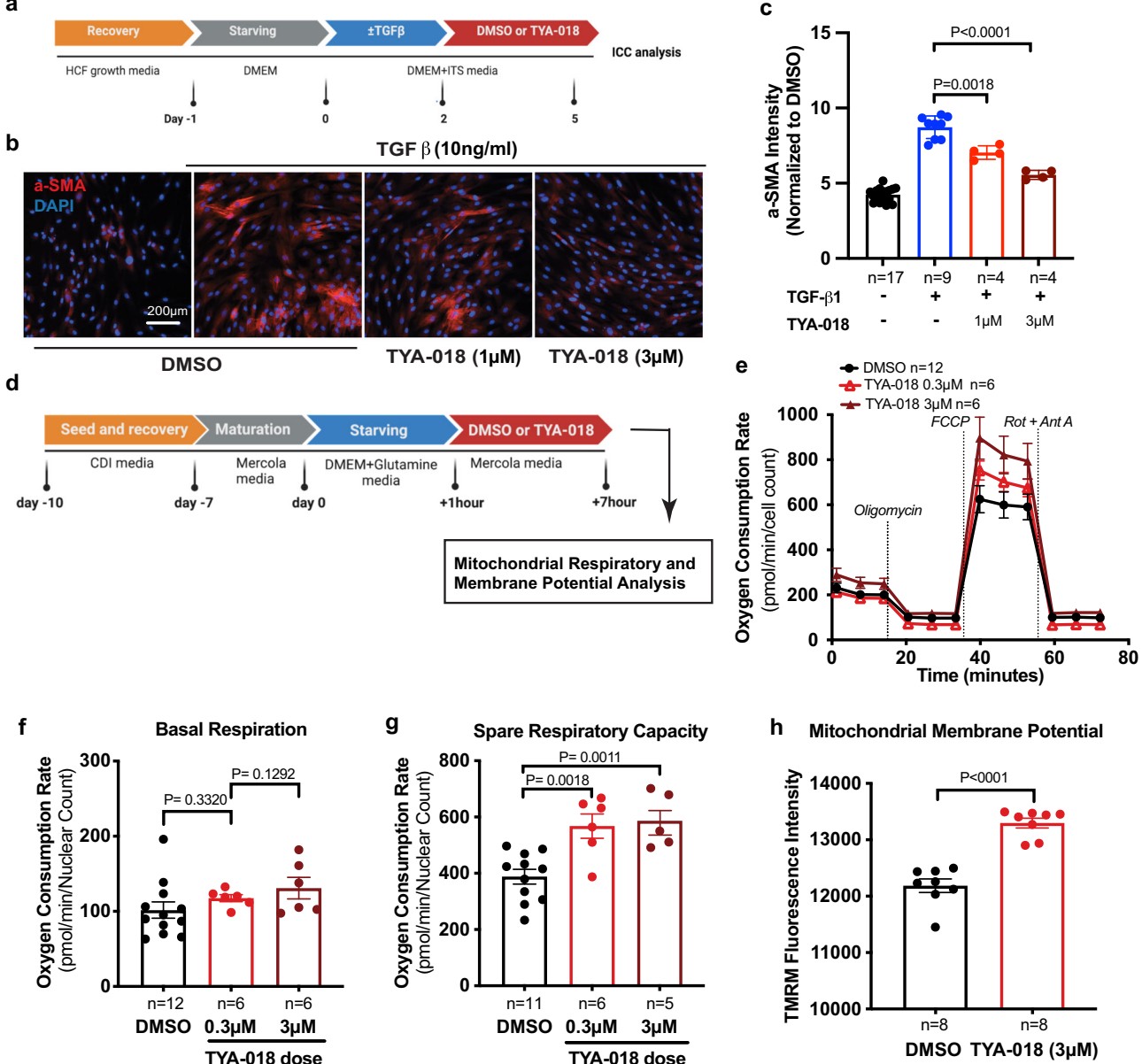

**Fig. 7 | TYA-018 reduces cardiac fibroblast activation and improves the metabolic signature of iPSC-CMs. a** Schematic diagram of an in vitro model of cardiac fibroblast activation using TGF-β. **b** Representative immunocytochemistry (ICC) images of α-SMA (red) and 4′,6-diamidino-2-phenylindole (DAPI; blue) staining in cardiac fibroblasts. Scale bar, 200 μm. **c** Quantification of cardiac fibroblasts positive for α-SMA after treatment with TYA-018 (1 and 3 μM). **d** Schematic workflow of Seahorse oximetry analysis in human iPSC-CMs. Seahorse oximetry kinetics (**e**), and quantification of basal respiration (**f**), reserve respiratory capacity (measured by the oxygen consumption rate) (**g**) in iPSC-CMs after treatment with TYA-018 0.3 μM and TYA-018 3 μM versus DMSO control. **h** Quantification of average florescent TMRM signal (normalized to DMSO) in iPSC-CMs treated with TYA-018 (3 μM) versus DMSO control. Data are expressed as the mean ± SEM. Statistical analysis was performed using two-tailed unpaired Student *t* test (**c**, **f–h**). The exact *P* values and replicates are shown in the figures. Source data are provided as a Source Data file.

HFpEF. More importantly, the combination therapy at low doses exhibited efficacy and offers potential benefits for translation into clinical practice, including a broader safety margin in clinical settings and increased patient accessibility and tolerability.

### HDAC6 inhibition reduces fibroblast activation
To determine whether TYA-018 directly affects cardiac fibroblasts, we developed an in vitro model in which we induced fibrosis in human cardiac fibroblasts with TGF-β. Consistent with RNA-Seq data from HFpEF mouse heart tissue, TYA-018 ameliorated cardiac fibroblast activation in vitro (Fig. 7a–c and Supplementary Fig. 8a–e). This finding suggests that HDAC6 inhibition directly affects fibroblasts.

### HDAC6 inhibition improves mitochondrial function
Next, we assessed whether TYA-018 directly affects the cell's metabolic state using human induced pluripotent stem cell–derived cardiomyocytes (iPSC-CMs). We measured the metabolic status and mitochondrial function of human iPSC-CMs treated with TYA-018 versus vehicle [dimethyl sulfoxide (DMSO)] (Fig. 7d). Seahorse oximetry analysis showed TYA-018-treated iPSC-CMs had a significantly higher reserve respiratory capacity than vehicle-treated iPSC-CMs, whereas both groups had similar basal respiration rate (Fig. 7e-g). We also measured mitochondrial membrane potential in iPSC-CMs using tetramethylrhodamine, methyl ester (TMRM). TMRM accumulates in the membrane of active mitochondria and is commonly used to indicate mitochondrial health[20]. TMRM analysis

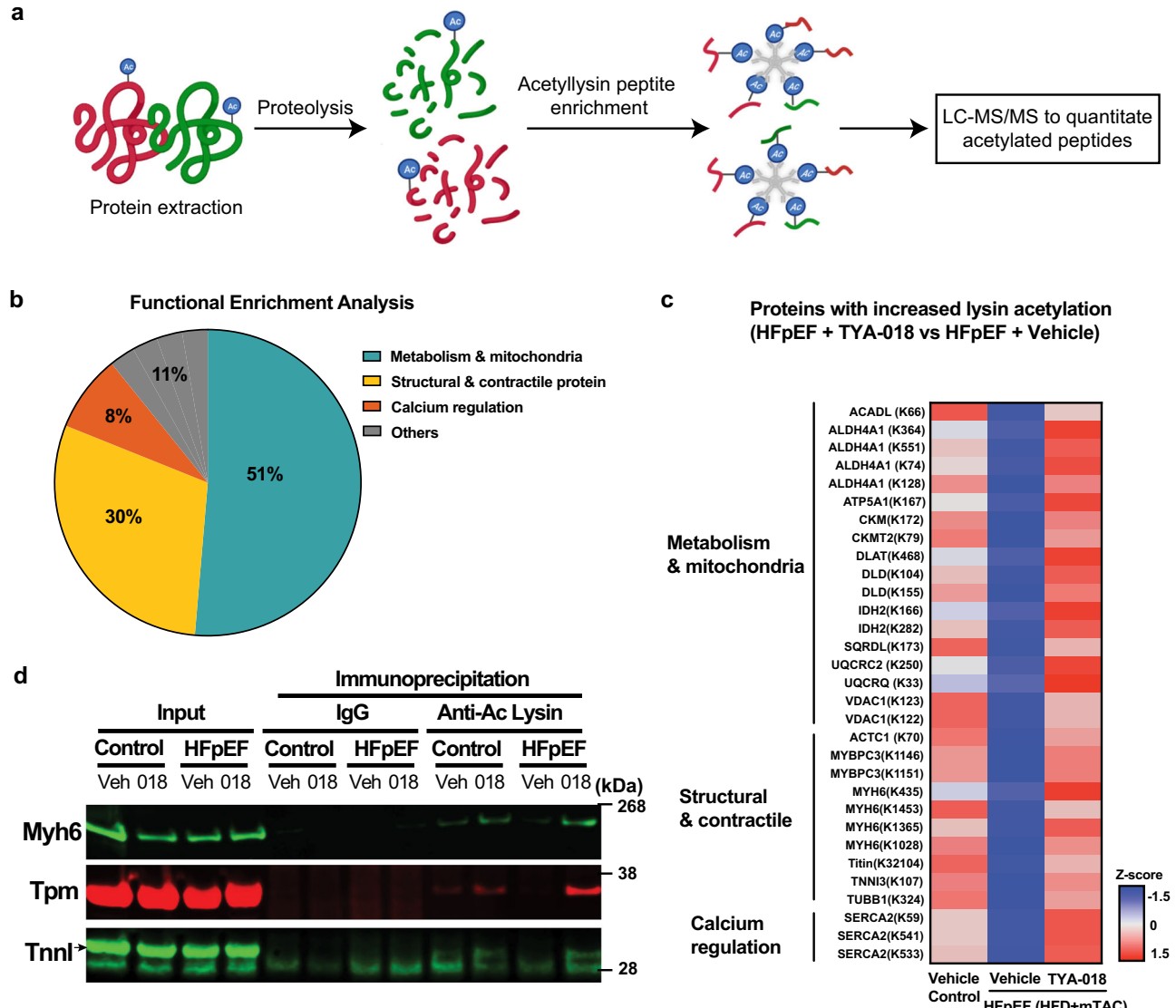

**Fig. 8 | TYA-018 regulates the acetylation of proteins associated with myofilaments, mitochondrial metabolic enzymes, and calcium regulation in HFpEF mice. a** Schematic overview illustrating acetylome workflow. **b** Pie chart shows STRING analysis of enriched proteins with altered acetylation in HFpEF mouse hearts after TYA-018 treatment. **c** Heatmap of altered lysine acetylation at sites of selected proteins after STRING analysis. Red represents relative increases in the lysine acetylation, and blue represents relative decreases in lysine acetylation. Proteins with their respective acetylation sites are listed on the left side of each row ($n = 3$ animals per group). **d** Representative western blot of tropomyosin (Tpm), troponin I (TnnI), and myosin heavy chain 6 (Myh6) after immunoprecipitation with anti-lysine acetylation antibody in heart samples. Immunoglobulin G (IgG) pulldown was used as a control.

showed that TYA-018 treatment increased mitochondrial membrane potential in iPSC-CMs (Fig. 7h). These results suggest that HDAC6 inhibition improves mitochondrial function in iPSC-CMs and augments their capacity to respond to an increased metabolic demand.

### HDAC6 inhibition targets metabolic and cardiac substrate
To identify potential substrates of HDAC6, we performed a comprehensive and unbiased acetylome analysis of heart tissue from vehicle-treated control mice, vehicle-treated HFD+mTAC mice, and TYA-018-treated HFD+mTAC mice. Heart tissues were collected 2 h after one dose of vehicle or TYA-018 (15 mg/kg) (Supplementary Fig. 9a). Quantitative proteomics was analyzed by liquid chromatography with tandem mass spectrometry (LC-MS/MS) following enrichment for acetylated peptides using an anti-acetyl-lysine antibody (Fig. 8a). Quantitative analysis of tubulin acetylation showed similar results to

those generated by Western blot (Supplementary Fig. 9b, c), indicating a robust acetylome assay.

To characterize the molecular function of proteins whose acetylation was affected by HDAC6 inhibition, we performed STRING (Search Tool for the Retrieval of Interacting Genes/Proteins) enrichment analysis. This analysis revealed that HDAC6 regulates acetylation on multiple proteins, often at multiple sites. These proteins also spanned several areas of cardiac biology that are critical for HFpEF pathophysiology, including metabolic enzymes and myofilament and calcium regulatory proteins (Fig. 8b, c). A pulldown assay confirmed that contractile proteins are target substrates for HDAC6. Indeed, TYA-018 increased acetylation of tropomyosin, α-myosin heavy chain, and troponin I (Fig. 8d). These data suggest that HDAC6 inhibition may improve diastolic function by directly modulating myofilament- and calcium-associated proteins.

## Discussion

In this study, we hypothesized that HDAC6 plays a role in heart structure remodeling and diastolic dysfunction in HFpEF. We tested this hypothesis in multiple models of HFpEF, including one that we developed by combining an HFD with mTAC to replicate key hemodynamic features of HFpEF in patients. To recapitulate the heterogeneity of human HFpEF, we also used a well-established model in which HFpEF was induced by combining an HFD with L-NAME treatment[13]. Using these two models, we found that inhibiting HDAC6 with TYA-018 reverses preexisting diastolic dysfunction through multiple pathways in the heart. These pathways are associated with fibrosis and mitochondrial dysfunction, which both contribute to HFpEF pathogenesis and closely correlate with disease severity. As a result of normalized end-diastolic pressure, TYA-018 treatment improved lung congestion and exercise capacity. We also confirmed that TYA-018 directly benefits cardiomyocytes and fibroblasts in heart tissues of HFpEF models due to multi-modal mechanisms related to improved systemic metabolism and inflammation. We also found that TYA-018 and the FDA-approved empagliflozin were comparable in efficacy for treating HFpEF and that their combination had an increased effect. Mice with genetic Hdac6 knockout delayed the progression of HFpEF and were resistant to TYA-018, demonstrating that the therapeutic benefit of TYA-018 is an on-target effect. The results support the theory that HDAC6 inhibition can reverse hypertrophy and diastolic dysfunction in models of HFpEF.

The lack of appropriate animal models has been a significant hindrance in developing effective treatments for HFpEF. Our study provides evidence that the HFD combined with mTAC model effectively simulates the heterogeneous profile of HFpEF in patients, making it a valuable tool for developing new therapies. Unlike the HFD + L-NAME model, which is based on obesity and activation of inducible nitric oxide synthase[13], the HFD+mTAC model more accurately simulates the cardio-metabolic profile of human HFpEF. The study using this model has revealed that cardiac fibroblast activation and metabolic stress play a critical role in the development of HFpEF. In other words, these findings suggest that targeting these cellular processes may provide an effective therapeutic approach for HFpEF patients. The mRNA transcriptome signatures and pathways observed in hearts from HFD+mTAC mice are also in line with the pathophysiology of experimental heart failure and extensive cardio-metabolic disease, which further support the validity of this model.

The use of HDAC inhibitors was found to be a potential strategy to treat HF in humans. Pan-HDAC inhibitors, which target all the HDACs, were shown to be cardioprotective in various animal models of diastolic dysfunction[9,10,25], but their use can lead to potential toxicity due to their non-selective nature and effect on global gene expression. A better approach is to use inhibitors that target specific HDACs, such as TYA-018, a high selective inhibitor of HDAC6. In this study, we found TYA-018 reversed deacetylation of substrates involved in metabolism, calcium, and structural and contractile regulation in HFpEF. These findings support previous reports that acetylation of cardiac troponin I and sarcoplasmic/endoplasmic reticulum $Ca^{2+}$ ATPase 2a can improve myofilament relaxation and calcium sensitivity[26,27]. These pathways are believed to play a role in the pathophysiology of HFpEF.

TYA-018 is a highly selective, orally available HDAC6 inhibitor that we developed as a potential therapy for dilated cardiomyopathy[11]. This molecule has been found to enhance energy production in HFpEF models by increasing the expression of targets associated with fatty acid metabolism, protein metabolism, and oxidative phosphorylation. This leads to an increase in overall energetics of the heart. In addition, TYA-018 has been found to block the upregulation of genes related to collagen formation and extracellular matrix remodeling in HFpEF mice and in agonist-mediated human cardiac fibroblasts in vitro. These data support findings that pan-HDAC inhibition by ITF2357/givinostat attenuated extracellular matrix expansion and suppressed cardiac

fibroblast activation in a mouse model of diastolic dysfunction[9]. These findings suggest that HDAC6 is the primary isoform involved in fibrosis remodeling and diastolic function.

It is noteworthy that our study's findings differ from the previously reported research on the role of HDAC6 in diastolic dysfunction. Lin et al. discovered that Hdac6 knockout mice had increased myofibril stiffness, resulting in exacerbated diastolic dysfunction in the UNX/DOCA model[28]. However, our study found that Hdac6-KO mice protected heart against stress of HFD + L-NAME, and established diastolic dysfunction in WT mice was rescued by HDAC6 inhibition with TYA-018. One possible explanation for this disparity is that the UNX/DOCA model only induced mild diastolic dysfunction, as indicated by a 5 mmHg EDP increase, while both models used in our study resulted in over a 10 mmHg increase. Furthermore, both of our models rely on prolonged high-fat diet resulting in weight gain and glucose intolerance, creating systemic metabolic stress, an important contributor to the pathophysiology of HFpEF in the clinical setting, while the UNX/DOCA model is primarily a hypertension model without a metabolic component. Another important distinction between our findings and those of Lin et al. is that myofibrils are composed of sarcomeric proteins but lack mitochondria and cytoplasmic components (including microtubules), two important sites of HDAC6 cellular function and, therefore, not an optimal system for studying HDAC6 inhibition. We also identified four lysine sites—K32104, K19868, K24707, and K31877—in titin that exhibited increased acetylation levels in HFpEF mice treated with TYA-018 compared to HFpEF animals receiving the vehicle treatment. This finding diverges from their study that identified two sites (K13013 and K13597) located near the PEVK element[28]. Notably, among these four sites, K32104 acetylation demonstrated a reduction in HFpEF mice, which was subsequently reversed and increased following TYA-018 treatment. The differential acetylation patterns observed in titin may, in part, elucidate the contrasting outcomes observed in different model systems. Thus, there are several significant differences between the experimental systems used and mechanistic findings in our studies and those reported in Lin et al. that can account for differing findings related to the role of HDAC6 in HFpEF.

The genetically constitutive deletion of HDAC6 resulted in a delayed development of diastolic dysfunction in mice subjected to 16 weeks with HFD and L-NAME treatment. This finding suggests that HDAC6 plays a crucial role in the pathophysiology of HFpEF, although it is likely not the sole factor involved. One hypothesis is that embryonic deletion of Hdac6 might be partially compensated for during development by other non-nuclear protein deacetylases, such as Sirtuin 2 (SIRT2) and SIRT3[29–31]. Importantly, TYA-018 effectively improved diastolic dysfunction in wild-type HFpEF mice, yet this effect was not observed in Hdac6 knockout mice. This indicates that TYA-018's ability to enhance cardiac remodeling and diastolic function is specifically associated with HDAC6 as an on-target effect. The genetic evidence aligns with medicinal chemistry data highlighting TYA-018's selectivity for HDAC6, demonstrating more than 1000-fold selectivity against other HDACs (HDAC1-11) in enzymatic assays[13]. While these findings emphasize its selectivity within the HDAC family, further investigation is required to uncover any potential non-HDAC off-target effects of this small molecule that could contribute to its pharmacological activity.

While HDAC6 is primarily localized to the cytosol, it is known for its involvement in the deacetylation of cytoplasmic substrates, such as alpha-tubulin and heat shock protein 90. In our study, we observed a significant impact on gene expression associated with inflammation and metabolism following HDAC6 inhibition in both acute and chronic treatments. These findings align with emerging research that underscores HDAC6's ability to indirectly influence gene expression by modulating signaling pathways, ultimately affecting nuclear gene regulation. HDAC6-mediated deacetylation of cytoplasmic proteins

may trigger intracellular signaling cascades that culminate in the activation or repression of transcription factors, thereby altering gene expression profiles[32].

HFpEF is characterized by impaired mitochondrial oxidative capacity and a rapid depletion of high-energy phosphate. In our study, TYA-018 treatment restored metabolic transcripts in HFpEF mice and improved mitochondrial respiratory capacity and membrane potential in iPSC-CMs. These findings are consistent with previous studies showing that pan-HDAC inhibition by resveratrol improved mitochondrial function and biogenesis in a mouse model of diabetic cardiomyopathy[33]. Resveratrol also activates SIRT1, leading to deacetylation of PGC-1α and increased mitochondrial biogenesis. Suberoylanilide hydroxamic acid has also been shown to protect ischemic myocardium in a mouse model of ischemia/reperfusion injury by maintaining mitochondrial homeostasis through PGC-1α-mediated mitochondrial biogenesis. Additionally, resveratrol enhances cardiac metabolism by activating peroxisome proliferator-activated receptor α, a key regulator of fatty acid metabolism.

Our data showed that empagliflozin alone and the combination of TYA-018 and empagliflozin had increased effects on diastolic function in HFpEF mice. Empagliflozin targets the SGLT2 receptor, which is primarily found in skeletal muscle and kidney, but not in the heart[34,35]. Despite the lack of definitive evidence of molecular targets in cardiac tissue, the study suggests that empagliflozin can directly influence cardiac function. The efficacy of empagliflozin in HFpEF may be explained by its ability to promote cellular housekeeping function through increasing autophagic flux. SGLT2 inhibitors like empagliflozin simultaneously increase the expression and activity of nutrient deprivation signaling pathways (such as AMPK, SIRT1, SIRT3, SIRT6, and PGC-1α) and decrease the activation of nutrient surplus signaling pathways (such as mTOR) in diverse tissues under stress[36]. These effects potentially contribute to the additive benefits of empagliflozin in combination with HDAC6 inhibition by TYA-018.

Our study provides important insights into the direct effects of HDAC6 inhibition by TYA-018 on heart diastolic function. However, it is worth noting that the drug also demonstrated systemic effects, as evidenced by its normalization of glucose levels after a single dose. This suggests that HDAC6 inhibition may have metabolic effects beyond the cardiovascular system, which could be relevant for the treatment of metabolic disorders. The potential systemic effects of the drug raise important questions about the mechanisms underlying its observed improvements in heart function. Specifically, it is unclear whether the improvements were solely due to the drug's direct effects on cardiac fibroblasts and cardiomyocytes, or whether the normalization of glucose levels also played a role. Future studies could investigate the potential metabolic effects of HDAC6 inhibition in animal models of metabolic disorders, as well as the mechanisms underlying these effects.

In conclusion, our study found that HDAC6 inhibition by TYA-018 reverses existing cardiac hypertrophy and diastolic dysfunction in multiple models of HFpEF, including the newly developed HFD+mTAC mouse model. Our findings show that HDAC6 inhibition involves multiple mechanisms in the heart, such as fibrosis, mitochondrial dysfunction, and systemic metabolism and inflammation, which highlight the multi-modal role of HDAC6 in HFpEF. We also discovered that TYA-018 has similar or superior efficacy compared to an FDA-approved drug, supporting the potential of targeting HDAC6 as a treatment for HFpEF in humans. Our study suggests that HDAC6 inhibition represents a promising new approach to address the unmet needs of HFpEF patients, either alone or in combination with current standard of care. Additionally, our results highlight the potential of TN-301, a chemical in the same series as TYA-018, which is currently in Phase 1 clinical testing and being developed for the treatment of HFpEF.

## Methods

### Ethical statement
Animal experiments were approved by the IACUC committee of Tenaya Therapeutics and were performed in compliance with the animal use protocol (AUP: 2020.005, 2020.007) and the National Institutes of Health. All ethical guidelines were adhered to whilst carrying out this study.

### Mice
Ten-week-old *Hdac6* knockout mice were purchased from The Jackson Laboratory (Strain #:029318). Only littermate males from breeding were studied. Ten-week-old male C57BL/6J (Cat. 000664) mice were purchased from The Jackson Laboratory (Bar Harbor, ME). Mice were acclimatised for 1 week before experiments. Mice were housed at 23–25 °C with light cycles of 12 h of light beginning at 6:00 a.m. and 12 h of dark beginning at 6:00 p.m., the humidity was 30–70%, and $H_2O$ was provided ad libitum. At the end of the study, mice were humanely euthanized in their home cage using acute exposure to $CO_2$ followed by rapid decapitation. Tissues were promptly collected and snap-frozen in liquid nitrogen for subsequent analysis.

### HFpEF mouse models
HFD+mTAC mice underwent mTAC surgery at 11 weeks old and were thereafter fed an HFD until euthanized. Another group of mice underwent mTAC and was fed standard chow (mTAC group). A third group did not undergo mTAC and was fed an high-fat diet (D12492 60 kcal% fat, Research Diets) (HFD group), and a fourth group did not undergo mTAC and was fed standard chow (D12450B 10 kcal% fat, Research Diets; chow group) (Fig. 1a, schematic overview of HFpEF model). HFD + L-NAME mice were generated as previously described[13]. One group of mice was fed an HFD, and another group was fed standard chow until euthanized.

### Modified trans-aortic constriction
The mTAC procedures were adapted from those previously described[37]. Mice were anesthetized with isoflurane (2–3%, inhalation) in an induction chamber. Then mice were intubated with a 20-gauge intravenous catheter and ventilated with a mouse ventilator (Minivent, Harvard Apparatus, Inc). Anesthesia was maintained with inhaled isoflurane (1–2%). A longitudinal 5-mm incision was made in the skin with scissors at the midline of the sternum. The chest cavity was opened by a small incision at the level of the second Intercostal space, 2–3 mm from the left sternal border. The chest retractor was gently inserted to spread the wound 4–5 mm in width. The transverse part of the aorta was bluntly dissected with curved forceps. Then, 6-0 silk was brought underneath the transverse aorta between the left common carotid artery and the brachiocephalic trunk. 24 gauge needle was placed directly above and parallel to the aorta. A loop was then tied around the aorta and needle and then secured with a second knot. The needle was immediately removed to create a lumen with a fixed stenotic diameter. The chest cavity was closed by 6-0 silk suture.

### Echocardiography and Doppler imaging
Transthoracic echocardiography was performed using a VisualSonics Vevo 2100 system equipped with a MS400 transducer (Visual Sonics, Toronto, ON). LVEF and other indices of systolic function were obtained from short-axis M-mode scans at the midventricular level, as indicated by papillary muscles in unconscious, gently restrained mice. Apical four-chamber views were obtained in anesthetized mice for diastolic function measurements using pulsed-wave and tissue Doppler imaging at the level of the mitral valve. Anesthesia was induced by 5% isoflurane and confirmed by lack of response to firm pressure on one of the hind paws. During echocardiogram acquisition, under body temperature–controlled conditions, isoflurane was reduced to 1.0–1.5% and adjusted to maintain a heart rate in the range of 450–500

beats per min. Parameters collected include: LVEF, LV mass, LVPWd, peak Doppler blood inflow velocity across the mitral valve during early diastole, peak Doppler blood inflow velocity across the mitral valve during late diastole, IVRT, and peak tissue Doppler of myocardial relaxation velocity at the mitral valve annulus during early diastole. At the end of the procedures, all mice recovered from anesthesia without difficulties.

## Tail-cuff blood pressure recordings
Systolic blood pressure was measured noninvasively in conscious mice using the tail-cuff method (BP-2000, Visitech Systems, Apex, NC). Mice were placed in individual holders on a temperature-controlled platform (37 °C) and recordings were performed under steady-state conditions. Before testing, all mice were trained to become accustomed to short-term restraint. Blood pressure was recorded for at least four consecutive days and readings were averaged from at least eight measurements per session.

## Intraperitoneal glucose-tolerance test
Intraperitoneal glucose-tolerance tests were performed by injecting glucose (2 g/kg in saline) after 6 h of fasting (from 7 a.m. to 1 p.m.). Tail blood glucose levels (mg/dL) were measured by AimStrip Plus blood glucose strips used with the AimStrip Plus blood glucose monitoring system (Germaine Laboratories, San Antonio, TX) at 0 (baseline, before dose glucose), 15, 30, 45, 60, and 120 min after glucose administration.

## Randomized efficacy test
After the HFpEF phenotypes were established with balanced parameters of echocardiography, HFD+mTAC or HFD + L-NAME mice were randomized to different groups (based on each study design) to receive oral doses of 15 mg/kg TYA-018, 10 mg/kg empagliflozin, or vehicle (5% DMSO + 45% PEG-300 + 50% purified water) at 10 mL/kg once per day for 6–9 weeks. TYA-018 was formulated in vehicle. Body weights were monitored during the study.

## Treadmill exercise stress test
A Ugo Basile Treadmill (cat# 47303) was angled at a 10° upward incline. Mice were run for 5 ms at a speed of 3 m/min, then for 3 min at 5 m/min. The speed was increased by 3 m/min every 2 min. At 16 min, a constant speed of 17 m/min was kept for the duration of the test. Mice were defined as reaching exhaustion when the mouse had two or more paws on the shock grid for 15 consecutive seconds. This treadmill test was repeated on each mouse for 7 days.

## Pressure-volume analysis
EDP was measured by pressure-volume loop analysis. The mice were intubated with a piece of polyethylene-90 tube and were placed on a warmed (37 °C) pad. The right carotid artery was then isolated. Care was taken to prevent damage to the vagal nerve. Mice were lightly anesthetized with isoflurane, and their heart rates were maintained rates at 450–550 beats per minute. A 1.2F Pressure-Volume Catheter (FTE-1212B-4518, Transonic, Inc.) was inserted into the right carotid artery and then advanced into the left ventricle. The transducer was advanced to the ventricular chamber, as evidenced by a change in pressure curves, and securely tied into place. The hemodynamic parameters were recorded in close-chest mode.

## Western blot analysis
Heart tissue was lysed in RIPA (radioimmunoprecipitation assay) buffer (25× v/w) and homogenized in a bead blender at the max intensity and 4 °C for 5 min. Samples were kept on ice for 30 min and then centrifuged at >15,000 g at 4 °C for 15 min. The supernatant was combined with Bolt LDS Sample Buffer and Bolt Sample Reducing Agent. Samples were loaded in the Bolt 4–12% Bis-Tris Gel and ran with the 1X Bolt MES SDS running buffer. Proteins were transferred onto nitrocellulose membranes using the iBlot 2 Gel Transfer Device. Nitrocellulose membranes were then blocked for 1 h at room temperature and then exposed to the primary antibody for 1 h at room temperature or overnight at 4 °C. Membranes were then washed with phosphate-buffered saline with 1% Tween 20 (PBS/T) three times for 5 min each at room temperature. Membranes were then exposed to secondary antibody for 1 h at room temperature, and then washed with PBS/T three times before being imaged with the LI-COR Odyssey image system. Primary antibodies included anti-HDAC6 (1:1000; Abcam, ab239362), anti-acetyl-a-tubulin (1:5000; Abcam, ab179484), anti-a-tubulin (1:5000; Abcam, ab7291), tropomyosin (1:1000; Abcam, ab133292), skeletal muscle myosin (1:1000; Santa Cruz, sc-32732), troponin I (1:1000; sc-133117), and anti-GAPDH (1:5000; Abcam, ab181602 and 1:5000; Themo Fisher, MA5-15738). All secondary antibodies were tagged with IRDye (LI-COR Biosciences).

## Pull-down assay
Heart tissue was lysed in T-PER Tissue Protein Extraction Reagent (20× v/w) and homogenized in a bead blender at the max intensity and 4 °C for 5 min. Samples were kept on ice for 30 min and then centrifuged at >15,000 g at 4 °C for 15 min. Then 300 μL of lysate was incubated with the acetylated lysine multitab rabbit monoclonal antibody (cat#: 9814, 1:50 dilution) overnight at 4 °C with rotation. Recombinant rabbit immunoglobulin G monoclonal antibody was used as an isotype control (cat#: ab172730). The Dynabeads Protein A Immunoprecipitation Kit (cat#: 10006D) was used for the pull-down. The Dynabeads were vortexed for >30 s to resuspend them in solution. Then 200 μL of the resuspended Dynabeads were transferred to a tube and placed on a magnet to remove the supernatant. Dynabeads were then washed with wash buffer twice and then the wash buffer was removed. The lysate/antibody mix was then placed with the Dynabeads and mixed at 4 °C with rotation for 1 h. This mix was then placed on a magnet, and the supernatant was removed. The Dynabeads were washed three times with the wash buffer and then the wash buffer was removed. Then 40 μL of elution buffer and 20 μL of LDS Sample Buffer/Sample Reducing Agent mix were added. The tubes were heated at 70 °C for 10 min and then placed on the magnet again. The supernatant was removed and loaded into gels for Western blot analysis.

## iPSC culture, cardiomyocyte differentiation, and purification
iCell Cardiomyocytes[2] were purchased from Fujifilm Cellular Dynamics, Inc. (Madison, WI).

## Seahorse oximetry
The Agilent Seahorse XFe96 Analyzer was used to measure mitochondrial function in iPSC-CMs. iCell Cardiomyocytes[2] were thawed according to the manufacturer's instructions (FUJIFILM Cellular Dynamics, Inc.) and placed directly onto Matrigel-coated Seahorse XF96 V3 PS Cell Culture Microplates (101085-004) at 15,000 cells/well. Once beating monolayers formed, the media was changed to Mercola maturation media[38] for 1 week to increase metabolic maturity of iPSC-CMs. Media was replenished every 3 days. The day of the assay, cells were incubated in starvation media containing 2 mM glutamine in Dulbecco's Modified Eagle Medium (DMEM) (Agilent, 103575-100) for 1 h. After starvation, cells were treated with TYA-018 (0.3 μM and 3 μM) in a final concentration of 0.1% DMSO diluted in Mercola Media for 6 h. Then, cells were washed and incubated with Seahorse XF DMEM Basal Medium supplemented with 2 mM glutamine, 1 mM pyruvate, and 3 mM glucose for 1 h. Then the Seahorse XFe96 cartridge was prepared according to the manufacturer's guidelines (Agilent Technologies). Basal oxygen consumption rates were measured followed by the Mito Stress Test (Agilent, 103015-100). Both assessments were done with inhibitors injected in the following order: oligomycin (2.5 μM), trifluoromethoxy carbonylcyanide phenylhydrazone (1 μM), rotenone, and antimycin A (0.5 μM). Oxygen consumption rates were normalized

to total nuclear count measured with Hoechst staining. Basal respiration was calculated as follows:

$$\text{Basal respiration} = \frac{\text{Last rate measurement before first oligomycin injection}}{\text{Minimum rate measurement after rotenone/antimycin}}$$

Reserve respiratory capacity was calculated as follows:

$$\text{Reserve respiratory capacity} = \frac{\text{Maximal Respiration after the addition of FFCP}}{\textit{Basal respiration}}$$

### TMRM, perchlorate assay

iCell Cardiomyocytes[2] were prepared as described for the Seahorse assay. Ten days after thaw, cells were exposed to a starvation media containing 2 mM glutamine in glucose-free DMEM (Gibco™, A1443001) for 1 h. After starvation, cells were treated with TYA-018 (3 µM) in a final concentration of 0.1% DMSO diluted in Mercola Media for 6 h. TMRM (Invitrogen™, T668) was prepared according to the manufacturer's instructions. Cells were treated with TMRM at a final concentration of 100 nM and incubated at 37 °C for 45 min. Fluorescence intensities were acquired at excitation/emission, and images were acquired in the 594 nm Texas Red filter channel using Cytation 5 (BioTek).

### RNA extraction and TaqMan qPCR analysis

Approximately $10^6$ cells or 20–30 mg of mouse heart tissue were lysed with TRI Reagent (Zymo Research) and frozen at −80 °C to ensure complete cell lysis. Total RNA was extracted and washed from lysed cells using the Direct-zol-96 RNA Kit (Zymo Research) according to the manufacturer's instructions. Samples were treated with DNase I for 15 min at room temperature. cDNA was reverse transcribed from ~1 µg of RNA through random hexamers using the SuperScript III First-Strand Synthesis System according to the manufacturer's instructions (Thermo Fisher Scientific). Real-time qPCR reactions were performed using the TaqMan universal PCR master mix (Thermo Fisher Scientific) with the TaqMan probes listed in Supplementary Table 1. RT-qPCR reactions were performed using the QuantStudio7 Flex Real-Time PCR systems (Thermo Fisher Scientific). Each reaction was performed in triplicate for a given RT-qPCR run, and each condition had four experimental replicates. Relative expression of the gene of interest was normalized to GAPDH as the housekeeping control using the 2−ΔΔCT method. *P* values were calculated with Student's *t* tests.

### Transcriptional analysis with RNA-Seq

From each replicate, 100 ng total RNA was extracted via the polyA-tail-specific protocol according to Illumina, Inc. The RNA libraries were prepared using a TruSeq Stranded mRNA kit (Illumina), which also removes ribosomal RNA. The libraries were sequenced as 50 base pair paired-end reads using Illumina NovaSeq V1.5 SP with an average of 99.86 million reads per sample. After adapter trimming by *fastp* (version 0.23.3)[39], raw RNA-Seq reads from mouse hearts in FASTQ format were aligned directly to the GENCODE (version M25) for reference transcript assembly (GRCm38.p6 and Ensembl 100) using *salmon* (version 1.6.0)[40] with best practice parameters to ensure mapping validity and reproducibility (--seqBias --gcBias --posBias --useVBOpt --rangeFactorizationBins 4 --validateMappings --mimicStrictBT2). Next, a script using *tximport* was used to generate an expression matrix normalized to transcripts per million. In this analysis, we only used genes detected in at least 90% of all samples. Protein-coding genes were determined using Ensembl release mus musculus annotations (GRCm38, Apr 2020) and extracted by *biomaRt* (version 2.46.3). Mitochondrial genes were also omitted, followed by renormalization to transcripts per million (TPM). All the analysis was performed on *Log2* transformed TPM + 1 values.

To evaluate functional perturbations, we performed pre-ranked using GSEA developed by the Broad Institute (version 4.1.0). GSEA assesses whether differences in expression of gene sets between two phenotypes are statistically significant[19]. Before analysis, a ranked list was calculated with each gene assigned a score and direction ("+" or "−") based on the *t*-statistics values. Gene sets were only considered statistically significant if the false discovery rate was less than 0.25 as determined with multiple hypothesis testing correction[41]. The normalized enrichment score, which reflects the degree to which a gene set is overrepresented in the ranked list and normalized for gene set size, was used to select significantly altered gene sets. The Pearson correlation coefficient was used to calculate the correlation between genes of interest and cardiac diastolic function parameters.

### Single nucleus RNA-seq analysis

Approximately 100 mg of mouse heart tissue was lysed with TRI Reagent (Zymo Research) and frozen at −80 °C to ensure complete cell lysis. Nuclei isolation was performed as per instructions from 10x Genomics "Nuclei Isolation from Cell Suspensions & Tissues for Single Cell RNA Sequencing" protocol (CG000124, Rev F) to reach the target concentration of 700–1200 nuclei/µl. The suspension was processed using 10X Genomics Single Cell protocol (CG00053, Rev D). We generated 3′ single-cell gene expression libraries (Next GEM v3.1) using the 10x Genomics Chromium system. Subsequently, Illumina NovaSeq (PE150) sequenced each library. For snRNA-seq libraries, sequencing utilized the HiSeq4000 platform at Singulomics (New York, NY, USA). Clean reads from sequencing were analyzed with Cell Ranger v6.1.2 software, aligning to the mouse reference genome mm10-2020-A.

The Cell Ranger output, comprising filtered gene expression matrix h5 files, underwent analysis using R package Seurat version 4. Genes expressed in fewer than three cells were filtered out. Cells were further filtered, retaining those expressing at least 250 genes, with a minimum of 1000 unique molecular identifiers (UMIs) and less than 10% mitochondrial gene content. Cells with UMI counts exceeding the average UMI count of all cells plus three times the standard deviation were removed.

Raw counts underwent natural log transformation and were scaled by a factor of 1000. A subset of 2000 genes, exhibiting the highest variance across all cells, was selected as variable features. Following regression against the number of UMIs per cell, the dataset was centered and scaled. Principal Component Analysis was conducted on the 2000 most variable genes. An elbow plot, depicting cumulative total variance explained against the number of Principal Components (PCs), indicated 50 as the optimal number for downstream analysis.

The Uniform Manifold Approximation and Projection method was employed to visualize cell representation in two-dimensional scatter plots, derived from the 50 identified PCs.

We employed a graph-based method for categorizing cells into distinct clusters. The process began with the construction of a k-nearest neighbor graph, which was based on the PCs selected earlier, using Euclidean distance as the metric. We then refined the edge weights between pairs of cells by assessing the Jaccard similarity, which reflects the shared overlap in their respective local neighborhoods. This refinement was carried out using the "FindNeighbors" function from the Seurat software package. Subsequently, we applied a modularity optimization technique, utilizing the Louvain algorithm, to progressively assemble cells into groups. This step aimed to maximize the standard modularity function and was executed via the "FindClusters" function in Seurat. The granularity of the resulting clusters was controlled by setting the resolution parameter at 0.6. For identifying cluster marker genes, we used the "FindAllMarkers" function for each cluster. These genes were then filtered based on specific criteria: a Bonferroni-corrected p-value of less than 0.05, a log2 fold

change greater than 0.25, and expression in a minimum of 25% of the cells within a cluster. We further compared these cluster marker genes against previously published markers to ascertain cell types. Lastly, differential expression analysis was conducted, and measures were considered significant if they fell below a false discovery rate threshold of 0.05.

To evaluate functional specification of each cluster, we performed pre-ranked using GSEA, as described earlier, on the Gene Ontology gene sets cataloged in MSigDB. Ranked list for each cluster was calculated by Seurat's FindMarkers using "DESeq2" test with each gene assigned a score and direction ("+" or "−") based on the minus log2 "P values" and "log2FoldChange", respectively. To achieve a comprehensive list of altered genes for each cluster, we relaxed the parameters to include genes altered in both directions (only.pos = FALSE) and set the following parameters: logfc.threshold = 0, min.pct = 0.05. Gene sets were only considered statistically significant if the false discovery rate was less than 0.25 as determined with multiple hypothesis testing correction. The normalized enrichment score was used to select significantly altered gene sets.

### Fibroblasts and in vitro fibrosis model

**Human cardiac fibroblast cultures.** Commercially available human cardiac fibroblasts (HCFs) (PromoCell, C-12375) were expanded for cell culture experiments with the Cardiac Fibroblast Growth Medium (CELL Application, INC, cat# 316-500). Briefly, HCFs were expanded in growth medium at 37 °C in a 5% $CO_2$ humidified incubator. At passage 3, HCFs were frozen at 3 million cells/vial using Cryostor (BioLifeSolutions, Car No. 20210). Freshly frozen vials of HCFs were used for every independent experiment.

**Human cardiac fibroblast activation.** HCFs were seeded with growth medium in 96-well plates (Greiner Bio-one) precoated with 0.1% gelatin. HCFs were serum starved for 24 h and then treated with 10 ng/ml TGF-β1 (R&D Systems™, cat# 77-54B-H100CF) in DMEM (Gibco, cat# 10566016) and 100X Insulin-Transferrin-Selenium (Thermo Fisher Scientific, cat# 41400045) for 48 h. Then, cells were washed with PBS and treated with vehicle (DMSO, cat# D2650-100 ml) or drugs for 72 h.

**Immunocytochemistry.** To detect activation of stress fibers in HCFs treated with TGF-β1, cells were fixed in 4% paraformaldehyde (Thermo Fisher Scientific, cat# PI28908) for 15 min, permeabilized in PBS/T for 15 min, and the blocked in 4% BSA + PBS/T solution for 1 h. Then cells were incubated overnight with primary antibody for α-SMA (1:500, Sigma, cat# A 2547) in 4% BSA + PBS/T and then with donkey anti-mouse Alexa Fluor Plus 594 (1:2000, Thermo Fisher Scientific, cat# A32744) secondary antibody. To visualize nuclei, cells were incubated with Hoechst stain (Thermo Fisher Scientific, cat# H3570) in PBS for 10 min. α-SMA and Hoechst staining were visualized with Cytation 5. The mean intensity of α-SMA staining and nuclear count was recorded for further analysis.

### Acetylome analysis

Mouse hearts were snap frozen in liquid nitrogen after harvest and then sent to Cell Signaling Technology for protein extraction and analysis. Heart extracts were sonicated, centrifuged, reduced with dithiothreitol, and alkylated with iodoacetamide. Then 5 mg of total protein for each heart was digested with trypsin, purified over C18 columns (Waters), and enriched using the PTMScan HS Acetyl-Lysine Motif Antibody (CST cat# 46784) as previously described[42]. Enriched samples were purified over C18 tips, and LC-MS/MS analysis was performed using an Orbitrap-Fusion Lumos Tribrid Mass Spectrometer as previously described[43], with replicate injections of each sample run non-sequentially. Briefly, peptides were separated using a 50 cm × 100 μM PicoFrit capillary column packed with C18 reversed-phase resin and eluted with a 90-min linear gradient of acetonitrile in 0.125%

formic acid delivered at 280 nL/min. Full MS parameter settings are available upon request.

MS spectra were evaluated by Cell Signaling Technology using Comet and the GFY-Core platform (Harvard University)[44]. Searches were performed against the most recent update of the National Center for Biotechnology Information *Mus musculus* database with a mass accuracy of ±20 ppm for precursor ions and 0.02 Da for product ions. Results were filtered to a 1% peptide-level false discovery rate with a mass accuracy of ±5 ppm on precursor ions and presence of an acetylated lysine residue. Site localization confidence was determined with AScore[45]. All quantitative results were generated with Skyline[46] to extract the integrated peak area of the corresponding peptide assignments. The accuracy of quantitative data was ensured by manual review in Skyline or in the ion chromatogram files. A two-tailed equal variance *t*-test was used to generate *P* values.

STRING enrichment analysis[47] was used to determine the molecular function of proteins with altered acetylation. For functional enrichment analysis, we included only proteins with changed acetylation in HFpEF (two folds change cutoff in HFpEF + vehicle vs control + vehicle; *P* < 0.05) and restored acetylation after HDAC6 inhibition (HFpEF+TYA-018 vs control + vehicle; *P* < 0.05).

### Statistics and reproducibility

All values are expressed as mean ± SEM. Statistical analyses were carried out by two-tailed unpaired Student *t* test for two groups or one- or two-way ANOVA followed by the Tukey post-hoc analysis for three or more groups. The statistical analyses used for each figure are indicated in the corresponding figure legends. Survival curves were plotted by the Kaplan–Meier method. Statistical analyses were conducted using Prism 10 (GraphPad Software). All experiments are represented by multiple biological replicates or independent experiments. The number of replicates per experiment is indicated in the legends or figures. All experiments were conducted using at least two independent experimental materials or cohorts to reproduce similar results. No sample was excluded from the analysis. A *p* value of less than 0.05 was considered significant.

The sample size was based on *n* values needed to evaluate differences between groups in prior studies. No statistical methods were used to predetermine the sample size. Whenever possible, group assignments were randomized and the investigators were blinded to allocation during experiments and outcome assessment.

### Reporting summary

Further information on research design is available in the Nature Portfolio Reporting Summary linked to this article.

## Data availability

Bulk RNA-Seq data and single nuclei RNA-Seq data that support the findings of this study have been deposited in Gene Expression Omnibus with the primary accession number GSE249414. The mass spectrometry proteomics data of acetylome analysis have been deposited to the ProteomeXchange Consortium via the PRIDE[48] partner repository with the dataset identifier PXD048363. Source data are provided with this paper. Any additional data is available upon request to the corresponding author (Jin Yang flairjinyang@gmail.com).

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

## Acknowledgements

We thank Dr. Crystal Herron of Redwood Ink, LLC, for editing the manuscript. We thank Dr. Joe Hill for suggestions on the HFD + L-NAME mouse model. We thank Dr. Julio Medina at R2M Pharma for compound synthesis. We thank members of Tenaya's vivarium facility, drug discovery, and bioinformatic teams for their technical assistance and comments on the manuscript.

## Author contributions

J.Y. and T.H. conceived the idea and designed the experiments. J.Y., S.R., X.S., J.R.P., G.A., M.A.M., and T.H. supervised the studies. A.Z., I.W., A.G., A.B., E.X., S.R., C.E.M., R.S., F.G., and J.Y. performed the experiments. J.Y., A.Z., A.G., S.R., and T.H. interpreted the results of experiments. S.R., R.S., M.K., and F.F. performed the bioinformatics analysis. A.Z., I.W., and J.Y. developed the HFD +mTAC mouse model. S.R. and A.Z. contributed equally. J.Y. and S.R. wrote the manuscript with support from all authors.

## Competing interests

The authors of this publication are employed by Tenaya Therapeutics and hold stock in the company. This potential conflict of interest has been thoroughly reviewed and managed by Tenaya Therapeutics. There are no other competing interests.
