## [Peer Review File · Nature Communications]

Targeting HDAC6 to treat heart failure with preserved ejection fraction in miceREVIEWER COMMENTS

Reviewer #1 (Remarks to the Author):

This is a well-written and interesting manuscript based on the work published by this group last year showing that inhibition of HDAC6 with TYA-018 was efficacious in a mouse genetic model of diastolic heart disease. The authors present data in this manuscript from 2 mouse models of HFpEF in which treatment with TYA-018 reversed hypertrophy and diastolic dysfunction. Mechanistically, the authors show that TYA-018 restores the expression level of genes associated with hypertrophy, fibrosis and mitochondrial energy production in their rodent HFpEF model.

The work is important given the paucity of therapeutics currently available to treat HFpEF. Experimental approaches are rigorous, and the data are convincing. Nonetheless, some issues need to be addressed:

- 1) The authors state "However, HDAC6 knockout mice showed a slower disease progression and presented with less severe HFpEF characteristics". The data do show slower disease progression but does not appear to show less severe hypertrophy and diastolic dysfunction at the endpoint of the study. There is no significance difference indicated between wt and HDAC6 KO at the endpoint of HFD+L NAME treatment presented in fig 4E-G or Extended Data Fig. 4 F-I. The author's need to modify their statement to reflect the data in figures or show that they are indeed significant. In addition, the authors state "Given that both HDAC6 knockout mice and their littermates developed HFpEF phenotypes but to different extents after 16 weeks on HFD+L-NAME". But their data show that they that they develop HFpEF at different rates not different extents. Again they either need to show significance in their data or modify their description to reflect the data.
- 2) Fig 8D shows that treatment with TYA-018 increased acetylation of tropomyosin, α -myosin heavy chain, and troponin I by pulldown. Fig 8C shows which lysine(s) are acetylated on α -myosin heavy chain, and troponin I. Why isn't the site of acetylation shown for tropomyosin??
- 3) The authors should comment on the previous cited works showing acetylation sites of troponin I and SERCA2, and if they are the same or different from what they found. E.g. Troponin I was acetylated on K132 in the cited paper.
- 4) HDAC6 KO work by Lin et al showed increased myofibril stiffness resulting in exacerbated diastolic dysfunction believed to be caused by titin's differential acetylation. This work also found differential acetylation of K2104 on titin. Is that the same or different Lys identified in the Lin et al study?
- 5) The authors compare their results to other studies using resveratrol which activates SIRT1 and SAHA which is a pan-HDAC inhibitor. In each case the inhibitors are affecting the activity of HDACs (SIRT1 or class I and IIa HDACs) which are known to be in the nucleus and can regulate gene expression by directly modifying the chromatin and transcription factors acetylation state. The authors need to discuss how inhibition of HDAC6 which is believed to be predominately in the cytosol might have such a dramatic impact on gene expression.
- 6) Further several of the proteins with increased lysine acetylation with HFpEF + TYA-018 shown in figure 8 C are in the matrix of the mitochondria. How does HDAC6 activity affect their acetylation state? The authors need to at least bring this up in the discussion.
- 7) The authors show HDAC6 expression levels in human hearts with HFpEF is significantly higher than in controls hearts (Fig. 2D). But the data also showed that 20-30% of the normal patients had HDAC6 expression levels above the mean of the HFpEF patients (some even higher than found in HFpEF patients). The authors should discuss this.
- 8) This manuscript uses a "a newly developed model that combines moderate trans-aortic constriction and high fat diet to mimic...HFpEF". Since it is a new model, the authors should present a Kaplan Meier curve to show survival using this model. In addition, the description of the

mouse TAC model states they use “one needle with customized size”. The authors need to give more detail to describe a new model of “moderate TAC” for others to be able to adopt this model in their own work. Was it a 26-gauge needle?

Reviewer #2 (Remarks to the Author):

These authors have tested a small molecule inhibitor of HDAC6 in two models of heart failure with preserved ejection fraction (HFpEF). Based on a comprehensive panel of phenotyping studies, they conclude that the molecule (TYA-018) antagonizes multiple features typical of HFpEF, notably diastolic dysfunction (which the authors emphasize).

The paper is written as a “drug study”. I submit, however, that a stronger case could be made for a study setting out to determine the role of HDAC6 in HFpEF, an open question with potential therapeutic relevance. I suggest the authors consider reworking their presentation to address this question (role of HDAC6 in HFpEF), using their small molecule as one of several approaches to address this.

Nothing is provided to discern the mechanism whereby HDAC6 inhibition purportedly mitigates HFpEF. Transcriptomic analyses, as presented in the paper, are suggestive – hypothesis generating – but the absence of mechanistic insight is a major weakness.

Whereas the animals across multiple treatment groups are characterized extensively, it is surprising that blood pressure is never measured. Drug-induced lowering of blood pressure would be a trivial explanation for most of the phenotypic changes reported. This is required.

I am puzzled by the emphasis on a new model of HFpEF which the authors repeatedly compare against a “well-established” model of HFD+L-NAME. They state, without supporting evidence, that their model of mTAC+HFD “more accurately simulates the cardio-metabolic profile of human HFpEF”. Furthermore, the fact that their mTAC+HFD model is marked by declines in ejection fraction at 16 weeks (e.g. fig 1d) is a substantial flaw, as very few patients with HFpEF transition to HFREF. These mice are already starting to do this at 16 weeks, indicating that this “new model of HFpEF” fails to mirror one of the major features of human HFpEF.

The authors measure HDAC6 protein abundance. However, it is surprising that HDAC6 enzymatic activity is never measured. Does this increase in HFpEF? Is enzymatic activity suppressed by TYA-018 in HFpEF?

To the investigators’ credit, they largely tested models of disease regression, the clinically relevant question, rather than what is often done, testing disease prevention. This is a strength.

HDAC6 has been implicated in several types of parenchymal fibrosis, such as IPF. This was not tested here, which seems an obvious deficiency.

M-mode echo images lack space and time stamps.

Reviewer #3 (Remarks to the Author):

In this manuscript, the authors found that using the HDAC6 specific inhibitor TYA-018 could improve heart failure with preserved ejection fraction in multiple mouse models. The proteomics work along with other experiments, while out of my expertise, are generally well designed and performed. The data interpretation and presentation are also clear and straightforward. My only minor concern is since the authors had already performed transcriptome and acetylome analysis, a whole proteome analysis may potentially add more insights towards the mechanical and pathophysiological role that HDAC6 play in HFpEF.

Reviewer #1:

This is a well-written and interesting manuscript based on the work published by this group last year showing that inhibition of HDAC6 with TYA-018 was efficacious in a mouse genetic model of diastolic heart disease. The authors present data in this manuscript from 2 mouse models of HFpEF in which treatment with TYA-018 reversed hypertrophy and diastolic dysfunction. Mechanistically, the authors show that TYA-018 restores the expression level of genes associated with hypertrophy, fibrosis and mitochondrial energy production in their rodent HFpEF model.

The work is important given the paucity of therapeutics currently available to treat HFpEF. Experimental approaches are rigorous, and the data are convincing. Nonetheless, some issues need to be addressed:

1) The authors state “However, HDAC6 knockout mice showed a slower disease progression and presented with less severe HFpEF characteristics”. The data do show slower disease progression but does not appear to show less severe hypertrophy and diastolic dysfunction at the endpoint of the study. There is no significance difference indicated between wt and HDAC6 KO at the endpoint of HFD+L NAME treatment presented in fig 4E-G or Extended Data Fig. 4 F-I. The author’s need to modify their statement to reflect the data in figures or show that they are indeed significant. In addition, the authors state “Given that both HDAC6 knockout mice and their littermates developed HFpEF phenotypes but to different extents after 16 weeks on HFD+L-NAME”. But their data show that they that they develop HFpEF at different rates not different extents. Again they either need to show significance in their data or modify their description to reflect the data.

Response: We have made the necessary adjustments to align the statements with the presented data. Additionally, we have included statistical information in the graphs. In the context of 8 weeks and 12 weeks with HFD+L-NAME induction, it is evident that *Hdac6* KO mice exhibited less severe hypertrophy and diastolic dysfunction, as indicated by reductions in LVPWd, LV Mass, IVRT, and E/e' (Fig 3. B-D and Source Raw Data). However, it's important to note that there were no statistically significant differences between WT and *Hdac6*-KO mice in these parameters at the 16-week endpoint. *Hdac6*-KO mice decelerated the progression of HFpEF. These findings support the hypothesis that HDAC6 plays a pivotal role in the development of HFpEF.

2) Fig 8D shows that treatment with TYA-018 increased acetylation of tropomyosin, α -myosin heavy chain, and troponin I by pulldown. Fig 8C shows which lysine(s) are acetylated on α -myosin heavy chain, and troponin I. Why isn't the site of acetylation shown for tropomyosin??

Response: We examined the acetylation status of 24 distinct lysine sites within tropomyosin, and found none of the sites had increased acetylation when comparing HFpEF+TYA-018 to HFpEF+Vehicle. It is important to note that in our acetylome analysis, we set a 2-fold absolute change in acetylation as the threshold for identifying altered sites, based on the average of three samples. Differences among these three samples could potentially explain why the tropomyosin sites were not considered as

altered in our acetylome analysis. However, when we looked at the acetylation level of tropomyosin in individual samples, we found that in TYA-018 treated HFpEF hearts, sites K251, K248, and K231 had increased acetylation in two of the three samples (see Extended Data Excel File, samples 312 and 313 in HFpEF-TYA018 group).

3) The authors should comment on the previous cited works showing acetylation sites of troponin I and SERCA2, and if they are the same or different from what they found. E.g. Troponin I was acetylated on K132 in the cited paper.

Response: Our acetylome data analysis identified five lysine sites in Troponin I including K107 (shown in Fig. 8C), K118, K121, K132, and K59. Apart from K107, none of other acetylation sites were altered in HFpEF mice or in response to TYA-018 treatment.

Prior research has indicated that acetylation of K514 and K492 in SERCA2a can impair cardiac function [1, 2]. In our study, we observed a significant decrease ($p < 0.05$ and $FC > 2$) in acetylation of K541, K543, K533, and K352 in vehicle treated HFpEF compared to control animals. Notably, this reduction was reversed to near-normal levels in TYA-018 treated HFpEF animals (Extended Data Excel File). Our data suggests that, acetylation status of these sites may be associated with SERCA2a function in HFpEF and can be restored by HDAC6 inhibition with TYA-018.

4) HDAC6 KO work by Lin et al showed increased myofibril stiffness resulting in exacerbated diastolic dysfunction believed to be caused by titin's differential acetylation. This work also found differential acetylation of K32104 on titin. Is that the same or different Lys identified in the Lin et al study?

Response: Lin et al. identified two sites (K13013 and K13597) located near the PEVK element. However, in our acetylome analysis, which encompassed over 200 different lysine sites in titin, these two sites were not identified. Our data indicated that only four lysine sites (K32104, K19868, K24707, and K31877) within titin, had increased acetylation in TYA-018 treated HFpEF mice compared to vehicle-treated HFpEF animals. Among these four sites, K32104 acetylation was reduced in HFpEF mice and was reversed back and increased in response TYA-018 treatment (Extended Data Excel File). The different acetylation sites identified in titin may partially explain the opposite outcome observed in different model systems.

5) The authors compare their results to other studies using resveratrol which activates SIRT1 and SAHA which is a pan-HDAC inhibitor. In each case the inhibitors are affecting the activity of HDACs (SIRT1 or class I and IIa HDACs) which are known to be in the nucleus and can regulate gene expression by directly modifying the chromatin and transcription factors acetylation state. The authors need to discuss how inhibition of HDAC6 which is believed to be predominately in the cytosol might have such a dramatic impact on gene expression.

Response: While HDAC6 is primarily localized to the cytosol, it is known for its involvement in the deacetylation of cytoplasmic substrates, such as alpha-tubulin and heat shock protein 90 (HSP90). In our study, we observed a significant impact on gene

expression following HDAC6 inhibition in both acute and chronic treatments. These findings align with emerging research that underscores HDAC6's ability to indirectly influence gene expression by modulating signaling pathways, ultimately affecting nuclear gene regulation. HDAC6-mediated deacetylation of cytoplasmic proteins may trigger intracellular signaling cascades that culminate in the activation or repression of transcription factors, thereby altering gene expression profiles [3]. For example, HDAC6 regulates β -catenin nuclear translocation and epidermal growth factor receptor trafficking [4]. Inhibition of HDAC6 blocks epidermal growth factor-induced β -catenin nuclear localization and decreases c-Myc expression, leading to the inhibition of epithelial cell proliferation.

6) Further several of the proteins with increased lysine acetylation with HFpEF + TYA-018 shown in figure 8 C are in the matrix of the mitochondria. How does HDAC6 activity affect their acetylation state? The authors need to at least bring this up in the discussion.

Response: While the precise mechanisms through which HDAC6 influences the acetylation of mitochondrial matrix proteins remain incompletely understood, several plausible mechanisms can be postulated. One possible avenue is that HDAC6 indirectly impacts mitochondrial acetylation by regulating acetylation patterns of cytoplasmic proteins. Alterations in cytoplasmic acetylation patterns, in turn, could exert downstream effects on mitochondrial protein transport and function. Additionally, HDAC6 may engage in interactions with other deacetylases or acetyltransferases that directly participate in the acetylation processes within the mitochondrial matrix. For instance, SIRT3, a member of the sirtuin family of NAD⁺-dependent protein deacetylases, has emerged as a key player in mitochondrial protein acetylation regulation, especially in the context of cardiac pathologies. SIRT3's enzymatic activity involves deacetylating multiple enzymes involved in mitochondrial metabolism. In situations where SIRT3 is absent or its activity is compromised, mitochondrial proteins tend to become hyperacetylated, leading to functional alterations that contribute to mitochondrial functional improvement [5].

7) The authors show HDAC6 expression levels in human hearts with HFpEF is significantly higher than in controls hearts (Fig. 2D). But the data also showed that 20-30% of the normal patients had HDAC6 expression levels above the mean of the HFpEF patients (some even higher than found in HFpEF patients). The authors should discuss this.

Response: We appreciate the reviewer's keen observation and the opportunity to address this point regarding the HDAC6 expression levels in our study. As our study aims to investigate the role of HDAC6 in HFpEF, the primary focus was to demonstrate a statistically significant difference in HDAC6 expression between HFpEF patients and control subjects. The significantly higher HDAC6 expression in HFpEF patients is consistent with our hypothesis that HDAC6 may play a crucial role in this condition.

However, it is important to highlight that a subset of individuals within the control group exhibited HDAC6 expression levels surpassing not only the mean HDAC6 levels of HFpEF patients but, in some cases, even exceeding the highest levels observed among HFpEF patients. This intriguing observation hints at the possibility that HDAC6 expression is not exclusively associated with the presence of HFpEF but may be influenced by additional factors or individual-specific conditions. Consequently, further investigations are imperative to elucidate the contributing factors behind this variability. Future research could encompass an exploration of the clinical and physiological attributes of normal patients exhibiting elevated HDAC6 expression, as well as an examination of potential genetic or environmental influences that might contribute to these variations.

8) This manuscript uses a “a newly developed model that combines moderate trans-aortic constriction and high fat diet to mimic...HFpEF”. Since it is a new model, the authors should present a Kaplan Meier curve to show survival using this model. In addition, the description of the mouse TAC model states they use “one needle with customized size”. The authors need to give more detail to describe a new model of “moderate TAC” for others to be able to adopt this model in their own work. Was it a 26-gauge needle?

Response: We have incorporated the Kaplan Meier curve into the model characterization (**Extended Data Fig. 1L**). In this context, it is worth noting that mice subjected to the HFD+mTAC treatment exhibited an approximate 20% mortality rate within a 16-week period, while no mortality was observed in the other groups. This observation is in line with the intriguing HFpEF phenotype exhibited by this specific group. As per your suggestion, we have included more detailed information regarding the induction of the mTAC+HFD model. A 24-gauge needle was utilized to induce pressure overload in the left ventricle, resulting in an increase of 10-20mmHg.

Reviewer #2 (Remarks to the Author):

These authors have tested a small molecule inhibitor of HDAC6 in two models of heart failure with preserved ejection fraction (HFpEF). Based on a comprehensive panel of phenotyping studies, they conclude that the molecule (TYA-018) antagonizes multiple features typical of HFpEF, notably diastolic dysfunction (which the authors emphasize).

The paper is written as a “drug study”. I submit, however, that a stronger case could be made for a study setting out to determine the role of HDAC6 in HFpEF, an open question with potential therapeutic relevance. I suggest the authors consider reworking their presentation to address this question (role of HDAC6 in HFpEF), using their small molecule as one of several approaches to address this.

Response: As suggested, we have revised the manuscript to center it around the role of HDAC6 in HFpEF.

Nothing is provided to discern the mechanism whereby HDAC6 inhibition purportedly

mitigates HFpEF. Transcriptomic analyses, as presented in the paper, are suggestive – hypothesis generating – but the absence of mechanistic insight is a major weakness. Whereas the animals across multiple treatment groups are characterized extensively, it is surprising that blood pressure is never measured. Drug-induced lowering of blood pressure would be a trivial explanation for most of the phenotypic changes reported. This is required.

Response: In our study, we performed noninvasive blood pressure measurements before and after treatment in conscious mice using the tail cuff method. Blood pressure determination revealed sustained hypertension in HFD+L-NAME, regardless of TYA-018 treatment, as well as in Empagliflozin-treated mice (Extended Data Fig. 5E, F, and G). These findings suggest that the observed efficacy in the study can be attributed to a direct effect on the heart rather than an indirect improvement through blood pressure reduction and a decrease in afterload.

I am puzzled by the emphasis on a new model of HFpEF which the authors repeatedly compare against a “well-established” model of HFD+L-NAME. They state, without supporting evidence, that their model of mTAC+HFD “more accurately simulates the cardio-metabolic profile of human HFpEF”. Furthermore, the fact that their mTAC+HFD model is marked by declines in ejection fraction at 16 weeks (e.g. fig 1d) is a substantial flaw, as very few patients with HFpEF transition to HFrEF. These mice are already starting to do this at 16 weeks, indicating that this “new model of HFpEF” fails to mirror one of the major features of human HFpEF.

Response: In Figure 1D, we applied HFD, mTAC alone, and a combination of both to potentially induce an HFpEF model. Mice that received the HFD+mTAC combination treatment presented diastolic dysfunction with preserved ejection fraction. However, mice subjected to mTAC alone exhibited a significant decline in ejection fraction after 16 weeks. Remarkably, the combination group (HFD+mTAC) not only maintained ejection fraction at the 16-week mark but also consistently upheld this preservation throughout the entire efficacy evaluation period, as illustrated in Figure 3B. Furthermore, we conducted gene profiling on heart samples from the HFD+mTAC group, revealing transcriptional signatures associated with human HFpEF (Fig. 2A). This observation not only emphasizes the sustainability of this model but also highlights its ability to recapitulate the major features of human HFpEF.

The authors measure HDAC6 protein abundance. However, it is surprising that HDAC6 enzymatic activity is never measured. Does this increase in HFpEF? Is enzymatic activity suppressed by TYA-018 in HFpEF?

Response: We attempted to assess HDAC6 enzymatic activity in addition to its protein levels by measuring the acetylation status of its well-established substrate, tubulin, in heart samples from the two HFpEF models (Extended Data Fig. 5M, N). Unexpectedly, the level of tubulin acetylation showed a relatively consistent pattern between the control group and the HFpEF group. This observation suggests that other factors, such

as tubulin acetyltransferases like ATAT1 and additional deacetylases, including SIRT2 and HDAC5 [6], might also play a role in regulating tubulin acetylation levels in HFpEF.

Furthermore, we assessed tubulin acetylation following TYA-018 treatment. TYA-018 significantly increased tubulin acetylation (Extended Data Fig. 5M, N). Given the compound's high specificity for HDAC6, this result strongly indicates an inhibition of HDAC6 enzymatic activity.

To the investigators' credit, they largely tested models of disease regression, the clinically relevant question, rather than what is often done, testing disease prevention. This is a strength.

Response: Thanks for the positive feedback. Considering the potent ability of the compound to reverse disease phenotypes, a chemical within the same series as TYA-018, presently undergoing Phase 1b clinical trials and being developed for the treatment of HFpEF, holds promise.

HDAC6 has been implicated in several types of parenchymal fibrosis, such as IPF. This was not tested here, which seems an obvious deficiency.

Response: Diastolic dysfunction in HFpEF is frequently associated with elevated interstitial cardiac fibrosis [7]. However, our analysis of left ventricular interstitial fibrosis, performed using Trichrome staining on whole cardiac cross-sections, did not reveal substantial collagen deposition in HFpEF mice when compared to the control group.

To gain further insight, we conducted an analysis of fibrosis marker gene expression using bulk RNA-seq. We observed an upregulation of gene clusters associated with ECM structural constituents (Extended Data Fig. 2A). Quantitative real-time polymerase chain reaction (qRT-PCR) further confirmed a significant increase in *Col3a1* expression in both HFpEF models (Extended Data Fig. 2B and H), and this upregulation was positively correlated with HDAC6 protein levels (Extended Data Fig. 2B and J).

Furthermore, our in vitro experiments demonstrated that the inhibition of HDAC6 with TYA-018 effectively reduced the activation of human cardiac fibroblasts and suppressed the expression of fibrotic genes (Extended Data Fig. 7B-E). Collectively, these findings suggest that HDAC6 may contribute to HFpEF pathogenesis, at least in part, through its involvement in fibroblast activation.

M-mode echo images lack space and time stamps.

Response: We have included M-mode echo images in the supplementary data, complete with both spatial and temporal stamps (Extended Data Fig. 1E and Extended Data Fig. 2I).

Reviewer #3 (Remarks to the Author):

In this manuscript, the authors found that using the HDAC6 specific inhibitor TYA-018 could improve heart failure with preserved ejection fraction in multiple mouse models. The proteomics work along with other experiments, while out of my expertise, are generally well designed and performed. The data interpretation and presentation are also clear and straightforward. My only minor concern is since the authors had already performed transcriptome and acetyome analysis, a whole proteome analysis may potentially add more insights towards the mechanical and pathophysiological role that HDAC6 play in HFpEF.

Response: Thanks for reviewer's overall positive feedback. As HDAC6 primarily resides in the cytosol, functioning as a deacetylase enzyme, our study has focus on acetyome analysis, as it offers greater relevance in identifying potential direct substrates. Bulk RNA-Seq is also employed to profile downstream gene expression alterations following the cascade of HDAC6 target substrate acetylation, while single nuclear RNA-Seq aims to pinpoint target cell types. While protein-level changes occur further downstream, we acknowledge the valuable suggestion made by the reviewer that proteome analysis could offer additional insights into the mechanical and pathophysiological role of HDAC6 in HFpEF, potentially revealing new therapeutic targets for this condition. We intend to incorporate this assay into our future studies.

Reference

1. Gorski, P.A., et al., *Role of SIRT1 in Modulating Acetylation of the Sarco-Endoplasmic Reticulum Ca(2+)-ATPase in Heart Failure*. *Circ Res*, 2019. **124**(9): p. e63-e80.
2. Gorski, P.A., et al., *Identification and Characterization of p300-Mediated Lysine Residues in Cardiac SERCA2a*. *Int J Mol Sci*, 2023. **24**(4).
3. Li, Y., D. Shin, and S.H. Kwon, *Histone deacetylase 6 plays a role as a distinct regulator of diverse cellular processes*. *FEBS J*, 2013. **280**(3): p. 775-93.
4. Li, Y., et al., *HDAC6 is required for epidermal growth factor-induced beta-catenin nuclear localization*. *J Biol Chem*, 2008. **283**(19): p. 12686-90.
5. Parodi-Rullan, R.M., X.R. Chapa-Dubocq, and S. Javadov, *Acetylation of Mitochondrial Proteins in the Heart: The Role of SIRT3*. *Front Physiol*, 2018. **9**: p. 1094.
6. Li, L. and X.J. Yang, *Tubulin acetylation: responsible enzymes, biological functions and human diseases*. *Cell Mol Life Sci*, 2015. **72**(22): p. 4237-55.
7. Zile, M.R., et al., *Myocardial stiffness in patients with heart failure and a preserved ejection fraction: contributions of collagen and titin*. *Circulation*, 2015. **131**(14): p. 1247-59.

REVIEWERS' COMMENTS

Reviewer #1 (Remarks to the Author):

The revised manuscript has satisfied any of the concerns or critique that I raised. In my opinion the manuscript should be accepted for publication.

Reviewer #2 (Remarks to the Author):

No new comments.

Reviewer #3 (Remarks to the Author):

The authors have addressed my comments and I have no other concerns.